# Microglial transglutaminase-2 drives myelination and myelin repair via GPR56/ADGRG1 in oligodendrocyte precursor cells

Stefanie Giera[1,2], Rong Luo[1,2], Yanqin Ying[1,2,3], Sarah D Ackerman[4†], Sung-Jin Jeong[1,2,5‡], Hannah M Stoveken[6], Christopher J Folts[1,2], Christina A Welsh[2], Gregory G Tall[6], Beth Stevens[2], Kelly R Monk[4§*], Xianhua Piao[1,2*]

[1]Division of Newborn Medicine, Department of Medicine, Children's Hospital and Harvard Medical School, Boston, United States; [2]Department of Neurology, F. M. Kirby Neurobiology Center, Children's Hospital and Harvard Medical School, Boston, United States; [3]Tongji Hospital, Tongji Medical College, Huazhong University of Science and Technology, Wuhan, PR China; [4]Department of Developmental Biology, Washington University School of Medicine, St. Louis, United States; [5]Department of Neural Development and Diseases, Korea Brain Research Institute (KBRI), Daegu, South Korea; [6]Department of Pharmacology, University of Michigan Medical Center, Ann Arbor, United States

*For correspondence:
monkk@wustl.edu (KRM);
Xianhua.Piao@childrens.harvard.edu (XP)

Present address: [†]Institute of Neuroscience, University of Oregon, Oregon, United States; [‡]Convergence Brain Research Department, Korea Brain Research Institute, Daegu, South Korea; [§]Vollum Institute, Oregon Health and Science University, Portland, United States

**Abstract** In the central nervous system (CNS), myelin formation and repair are regulated by oligodendrocyte (OL) lineage cells, which sense and integrate signals from their environment, including from other glial cells and the extracellular matrix (ECM). The signaling pathways that coordinate this complex communication, however, remain poorly understood. The adhesion G protein-coupled receptor ADGRG1 (also known as GPR56) is an evolutionarily conserved regulator of OL development in humans, mice, and zebrafish, although its activating ligand for OL lineage cells is unknown. Here, we report that microglia-derived transglutaminase-2 (TG2) signals to ADGRG1 on OL precursor cells (OPCs) in the presence of the ECM protein laminin and that TG2/laminin-dependent activation of ADGRG1 promotes OPC proliferation. Signaling by TG2/laminin to ADGRG1 on OPCs additionally improves remyelination in two murine models of demyelination. These findings identify a novel glia-to-glia signaling pathway that promotes myelin formation and repair, and suggest new strategies to enhance remyelination.

DOI: https://doi.org/10.7554/eLife.33385.001

## Introduction

Myelination of axons is essential for both efficient impulse conduction and for health of nerve fibers. During the development of the vertebrate central nervous system (CNS), myelin is produced and maintained by oligodendrocytes (OLs), which arise from a lineage-restricted, proliferative pool of OL precursor cells (OPCs). OPCs are abundant in the adult CNS, generating new oligodendrocytes and new myelin under conditions of myelin damage, as is seen in demyelinating diseases and in rodent models of demyelination. Unfortunately, the regenerative capacity of OPCs after injury is poor, necessitating further investigation into the molecular mechanisms that guide both developmental and post-injury induced OPC differentiation. A number of studies have identified cell-autonomous regulators of OPC proliferation and terminal differentiation into myelinating OLs, but our

understanding of these processes remains incomplete (*Emery, 2010*; *Nave and Werner, 2014*; *Hughes and Appel, 2016*; *Mitew et al., 2014*). Furthermore, a comprehensive portrait that incorporates both cell-autonomous and -non-autonomous factors, including OPC interactions with other cell types and with extracellular matrix (ECM) proteins, is presently lacking (*Mayoral and Chan, 2016*; *Colognato and Tzvetanova, 2011*; *Clemente et al., 2013*; *Wheeler and Fuss, 2016*).

Cell-cell interactions with other glial populations, in particular microglia, influence OPC proliferation and maturation (*Hagemeyer et al., 2017*). While our understanding of the molecular mechanisms underlying OPC-microglial interactions during developmental myelination is limited, recent studies have offered some insight. For example, the nonspecific suppression of microglial function with the anti-inflammatory drug minocycline leads to a reduction in the numbers of neural progenitors and OPCs during postnatal CNS development, suggesting that a cell-cell interaction is critical to the maturation of the OL-lineage (*Shigemoto-Mogami et al., 2014*). By comparison, crosstalk between OPCs and microglia has been widely reported in the context of demyelinating insults (*Miron et al., 2013*), including in demyelinating diseases such as toxin-induced murine models of demyelination in which clearance of damaged myelin by microglia is necessary for OPC-dependent remyelination (*Kotter et al., 2006*). The molecular signaling pathways that mediate communication between microglia and oligodendrocyte lineage cells during development and repair, however, have not been fully delineated.

Similar gaps in knowledge exist for OPC communication with the ECM. The role of ECM proteins in modulating OPC proliferation and differentiation has been described (*Jagielska et al., 2012*; *Urbanski et al., 2016*). Nevertheless, the mechanisms through which OPCs communicate with the ECM and how these interactions are transduced into biologically- or therapeutically-relevant OPC behaviors remains to be elucidated.

To address the question of how OPCs integrate signals from the matrix and other glial cells, we focused on the family of adhesion G protein-coupled receptors (aGPCRs). Most aGPCRs possess a GPCR-Autoproteolysis-INducing (GAIN) domain that mediates autoproteolytic processing during protein maturation to generate an N- and a C-terminal fragment (NTF and CTF, respectively), which remain non-covalently associated on the cell surface (*Langenhan et al., 2013*; *Hamann et al., 2015*; *Langenhan et al., 2016*). These structural features make aGPCRs plausible transducers for cellular responses to ECM interactions, and specific aGPCRs have emerged as mechanical receptors and as transducers of cell-matrix interactions (*Scholz et al., 2015*; *Petersen et al., 2015*; *Langenhan et al., 2016*). Prior work has defined functions for several aGPCRs in the development of myelinating glial cells in both the peripheral and central nervous systems (*Monk et al., 2009*; *Petersen et al., 2015*; *Ackerman et al., 2015*; *Giera et al., 2015*; *Langenhan et al., 2016*; *Shin et al., 2013*; *Ackerman et al., 2018*). In particular, the aGPCR ADGRG1 (also known as GPR56) has been shown to regulate OL development and CNS myelination (*Ackerman et al., 2015*; *Giera et al., 2015*; *Salzman et al., 2016*).

ADGRG1 is an evolutionarily conserved regulator of OL development in zebrafish, mice, and humans (*Ackerman et al., 2015*; *Giera et al., 2015*). Loss-of-function mutations in *ADGRG1* cause the devastating human brain malformation called bilateral frontoparietal polymicrogyria (BFPP), which is comprised of a constellation of structural brain defects including CNS hypomyelination (*Piao et al., 2004*; *Piao et al., 2005*). Conditional deletion of *Adgrg1* in OL lineage cells results in CNS hypomyelination, and this is specifically caused by deficiencies in ADGRG1 signaling in OPCs and immature OLs (*Giera et al., 2015*). Loss of *Adgrg1* in mice and zebrafish decreases OPC proliferation, thereby leading to a reduced number of mature myelinating OLs and fewer myelinated axons in the CNS (*Ackerman et al., 2015*; *Giera et al., 2015*). However, the relevant ADGRG1 ligand during CNS myelination has not yet been defined.

In this study, we demonstrate that microglia-derived transglutaminase 2 (encoded by *Tgm2*) is a novel ADGRG1 ligand for OPCs. Microglia-specific deletion of *Tgm2* leads to reduced OPC cell division, fewer mature OLs, and hypomyelination during postnatal CNS development, phenocopying the loss of *Adgrg1*. *In vitro*, we find that the activation of ADGRG1 by TG2 requires the ECM protein laminin to promote OPC proliferation. Moreover, we extend our analysis of the role ADGRG1-TG2 interactions in developmental myelination to mouse models of myelin damage. We provide evidence that OPC-specific deletion of *Adgrg1* impairs CNS remyelination after toxin-induced demyelination, and that recombinant TG2 rescues remyelination failure in organotypic cerebellar slices in a ADGRG1-dependent manner. Taken together, these findings show that the tripartite signaling

complex comprised of microglial TG2, extracellular laminin, and OPC ADGRG1 regulates OL development and myelin repair.

## Results

### Putative ligands of ADGRG1 are expressed in microglia

aGPCR ligand binding is solely mediated by its N-terminal fragment (NTF)(*Hamann et al., 2015*). To establish the distribution of an OPC-specific ADGRG1 ligand in the developing brain, we labeled putative ADGRG1 binding proteins in mouse corpus callosum (CC) tissue with a probe comprised of the ADGRG1 NTF fused to human immunoglobulin Fc fragment (ADGRG1$^N$-hFc; *Figure 1A*). These studies were performed at postnatal day 5 (P5) when OPCs are actively proliferating. To identify the lineage of ADGRG1 ligand-expressing cells, we performed a series of double IHC experiments with ADGRG1$^N$-hFc paired with Iba1 (to label microglia), GFAP (to label astrocytes), and PDGFRα (to label OPCs). We observed robust and consistent putative ligand detection in microglia (*Figure 1B and C*), while no obvious putative ligand binding was detected in OPCs and only sparse signals were observed in astrocytes (*Figure 1D–G*). Quantitative analyses showed that ~80% Iba1 +microglia and~20% of GFAP +astrocytes express the putative ligand of ADGRG1 (*Figure 1H*).

To identify the ligand of ADGRG1 in OPCs, we employed an in vitro proteomics approach. Biotinylated ADGRG1$^N$-hFc was incubated with protein lysates isolated from cultured primary glial cells that were enriched for microglia and astrocytes, and streptavidin beads were used to capture biotinylated ADGRG1 and its glial-derived candidate binding partner(s) (*Figure 1—figure supplement 1*). Purified complexes were then submitted for identification by mass spectrometry following digestion by trypsin. Proteins with more than five identified tryptic peptides from three independent experiments are listed in *Table 1*.

### *Tgm2* knockout mice display reduced myelination

We first focused our attention on TG2 because it is a known binding partner of ADGRG1 in melanoma cells (*Xu et al., 2006*). To evaluate the candidacy of TG2 as the ligand of ADGRG1 in OPCs, we investigated whether deleting *Tgm2* phenocopies the myelination phenotype observed in *Adgrg1* knockout mice. Indeed, we observed significantly reduced numbers of *Plp*$^+$ OLs in the CC of *Tgm2* knockout mice at both P14 and P28 compared to littermate controls (*Figure 2A and B*), with reductions in *Plp*$^+$ OLs that are similar to reductions observed in *Adgrg1* knockout mice (*Giera et al., 2015*). Similar to our analysis of callosal *Plp*$^+$ OL density in adult *Adgrg1*-deficient mice (*Giera et al., 2015*) and consistent with a previous analysis of two-month-old *Tgm2*-deficient mice (*Van Strien et al., 2011*), four- to five-month-old *Tgm2* knockout mice showed normal levels of *Plp*$^+$ OLs (*Figure 2—figure supplement 1*), indicating a gradual correction of OL numbers through adulthood.

Transmission electron microscopy of P28 CC showed significantly fewer myelinated axons in *Tgm2*$^{-/-}$ mice compared to littermate controls (*Figure 2C and D*), whereas myelin thickness corrected for axon caliber (g-ratio), total axon numbers, and axon diameters were comparable between the two groups of animals (*Figure 2E*, *Figure 2-figure supplement 2*). A gene dosage effect was not apparent, as we detected no difference in the density of *Plp*$^+$ OLs or the percentage of myelinated axons in CC between *Tgm2*$^{+/+}$ and *Tgm2*$^{+/-}$ mice (*Figure 2—figure supplement 3*). Taken together, these data support the finding that TG2 is the ligand of ADGRG1 in OPCs.

### Microglia-specific ablation of *Tgm2* reduces OL numbers

Recent transcriptomic studies revealed that *Tgm2* is primarily expressed by microglia in postnatal murine CNS cells (*Zhang et al., 2014*), with expression detected in purified embryonic, early postnatal, and adult microglia (*Matcovitch-Natan et al., 2016*) (*Figure 2—figure supplement 4*). Consistent with these reports, our western blot analysis of freshly isolated, purified murine glial cells demonstrated selective enrichment in microglia, with no detectable TG2 protein in wildtype OPCs, OLs, astrocytes, or *Tgm2*$^{-/-}$ microglia (*Figure 2F*).

To test if microglial TG2 regulates developmental myelination, we studied the effect of conditionally deleting microglial *Tgm2* on the production of mature OLs. *Tgm2* floxed mice (*Nanda et al., 2001*) were crossed with *Cx3cr1*-Cre mice (*Yona et al., 2013*) where Cre recombinase expression is

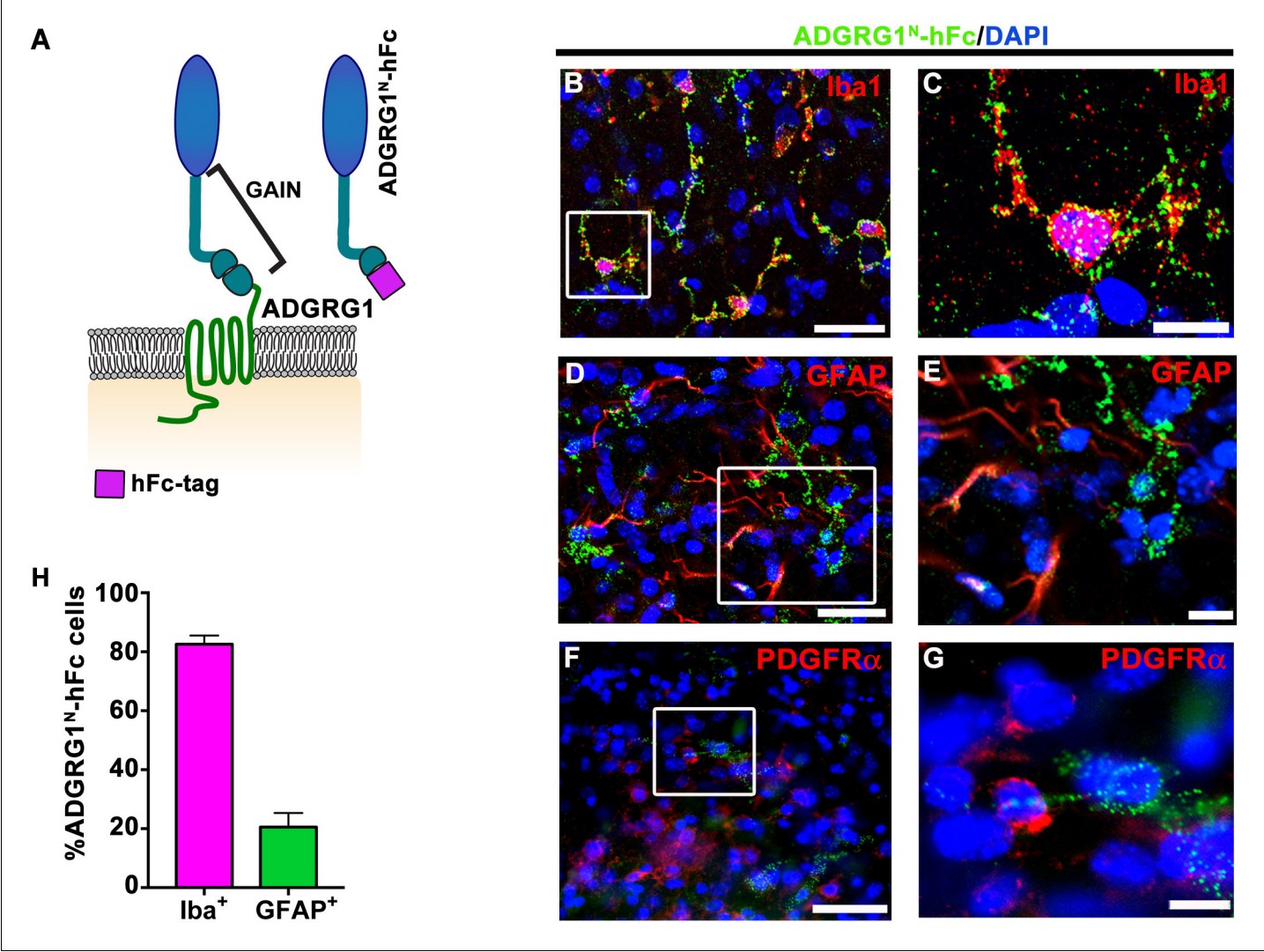

**Figure 1.** Microglia express the putative ligand of ADGRG1. (**A**) Schema of ADGRG1 receptor. GAIN domain and ADGRG1$^N$-hFc are shown. (**B**, **D**, and **F**) Double labeling of ADGRG1$^N$-hFc (green) and Iba1 (**B**, red), GFAP (**D**, red), PDGFR$\alpha$ (**F**, red) in P5 wt corpus callosum. DAPI, blue. Scale bar, 25 μm. (**C**, **E**, and **G**) Higher magnification of the boxed region in (**B**, **D** and **F**). Scale bar, 10 μm. Staining was repeated $N$ = 3–4 animals. (**H**) Quantification of double positive ADGRG1$^N$-hFc$^+$ and Iba$^+$ cells and ADGRG1$^N$-hFc$^+$ and GFAP$^+$ cells. $N$ = 3–4 per staining.

DOI: https://doi.org/10.7554/eLife.33385.002

The following figure supplement is available for figure 1:

**Figure supplement 1.** Flow chart of the in vitro biotinylation/proteomic approach.

DOI: https://doi.org/10.7554/eLife.33385.003

restricted to microglia. The numbers of *Plp*$^+$ OLs in the CC of microglia-specific *Tgm2* knockout mice (*Tgm2*$^{fl/fl}$;*Cx3cr1Cre*$^+$) were significantly reduced compared to littermate *Tgm2*$^{fl/+}$;*Cx3cr1Cre*$^-$ controls (***Figure 2G and H***). Importantly, the reduction is comparable to what was observed in the constitutive *Tgm2* knockout at P28 (***Figure 2A and B***), supporting the hypothesis that microglia are the primary source of TG2 during OL development.

## TG2 promotes OPC proliferation and cell cycle progression

Next, we investigated whether TG2 regulates OPC proliferation in the developing white matter, a function that has been previously documented for ADGRG1. Indeed, numbers of *Pdgfra*$^+$ OPCs in the CC of *Tgm2*$^{-/-}$ mice were significantly reduced compared to controls at P14 (***Figure 3A and B***). Our previous studies showed that loss of *Adgrg1* in the OL lineage causes premature cell cycle exit,

**Table 1.** Summary of mass spectrometry results from three independent purifications.
Total number of peptides sequences in each individual experiment specific for binding to ADGRG1$^N$.

| Protein name (Gene ID) | Peptide number (Exp1 + Exp2+Exp3) |
|---|---|
| Plectin (*Plec*) | 21 + 22 + 23 |
| Transglutaminase 2 (*Tgm2*) | 19 + 20 + 17 |
| GPR56/ADGRG1 (*Gpr56/Adgrg1*) | 12 + 11 + 9 |
| Growth arrest-specific 6 (*Gas6*) | 11 + 9 + 11 |
| Sodium/potassium-transporting ATPase subunit alpha-1(*Atp1a1*) | 8 + 7 + 9 |
| Aspartyl-tRNA synthetase (*Dars*) | 8 + 3 + 8 |
| Membrane protein, palmitoylated 6 (*Mpp6*) | 8 + 5 + 11 |
| Microtubule-actin crosslinking factor 1 (Macf1) | 8 + 13 + 8 |

DOI: https://doi.org/10.7554/eLife.33385.004

resulting in a diminished size of *Pdgfra*$^+$ OPC pools and fewer mature OLs (*Giera et al., 2015*). Therefore, we performed cell cycle exit assays (*Figure 3C*) and found significantly fewer BrdU$^+$/Ki67$^+$ double positive cells in the CC of *Tgm2* knockouts compared to the controls (*Figure 3D and E*). Consistent with decreased OPC proliferation, we also observed an overall reduction in BrdU$^+$ cells (*Figure 3F*) and fewer PDGFRα$^+$/BrdU$^+$ OPCs in *Tgm2* knockout mice compared to littermate controls (*Figure 3G and H*). We conclude that OPCs prematurely exit the cell cycle in the absence of TG2, as we previously reported in *Adgrg1*$^{-/-}$ mice (*Giera et al., 2015*). Taken together, our data support a model in which microglial-derived TG2 promotes OPC proliferation via ADGRG1.

## Other candidate ligands do not phenocopy loss of *Adgrg1*

We returned to our list of putative ADGRG1 ligands and asked if other candidates phenocopied the ablation of *Adgrg1*. Given that ADGRG1, like many aGPCRs, mediates cell-cell and cell-ECM interactions (*Hamann et al., 2015*; *Langenhan et al., 2013*), we prioritized the candidate binding partners of ADGRG1 by focusing on plasma membrane-associated and ECM proteins, and eliminated *Atp1a1*, *Dars*, and *Macf1*. Using zebrafish – a well-established model for studying myelination (*Preston and Macklin, 2015*; *Ackerman and Monk, 2016*), in which major classes of glia are present (*Lyons and Talbot, 2014*) and in which *Adgrg1* function in OL development is conserved (*Ackerman et al., 2015*) – we screened the remaining candidates. We generated new mutant alleles of genes encoding the zebrafish orthologs of Gas6 (*gas6*), Mpp6 (*mpp6a, mpp6b*), and Plectin (*pleca, plecb*) using CRISPR/Cas9 genome editing (*Figure 3—figure supplement 1A–E*). Whole mount in situ hybridization (WISH) for *myelin basic protein* (*mbp*) at 5 days post-fertilization did not reveal CNS myelination phenotypes in any of the putative-ligand zebrafish mutants at this stage (*Figure 3—figure supplement 1F–O*), a time point at which *adgrg1* zebrafish mutants exhibit reduced *mbp* expression (*Ackerman et al., 2015*). This lack of phenocopy suggested that Gas6, Mpp6, and Plectin do not function as ligands for ADGRG1 during OPC development. ADGRG1$^N$ was used as the bait for the pull-down assay; therefore, it is not surprising that ADGRG1 was detected in the MS analysis. Additionally, ADGRG1 has been reported to undergo trans-trans homophilic interactions (*i. e.*, interactions between ADGRG1 NTFs expressed on adjacent cells) (*Paavola et al., 2011*) and so it is also possible that ADGRG1$^N$-hFc probe immunoprecipitated ADGRG1 itself.

## TG2 functions together with laminin to stimulate OPC proliferation

To establish conditions for mechanistic investigation of the function of TG2 on OPC ADGRG1 during myelination, we performed in vitro analyses of wild-type (wt) and *Adgrg1*$^{-/-}$ OPC cell expansion in the presence of recombinant TG2 (rTG2). To our surprise, rTG2 failed to enhance the proliferation of wt OPCs grown on poly-D-lysine (*Figure 3I*). TG2 is present in the extracellular milieu and is capable of cross-linking ECM proteins, including laminin-111 (*Aeschlimann et al., 1992*; *Belkin, 2011*). In light of this interaction, and the observation that laminins are broadly detected in the postnatal CC and in close proximity with OPCs in vivo (*Relucio et al., 2012*), we tested the need for laminin-111 in promoting OPC proliferation in a ADGRG1- and TG2-dependent manner. We found that OPC

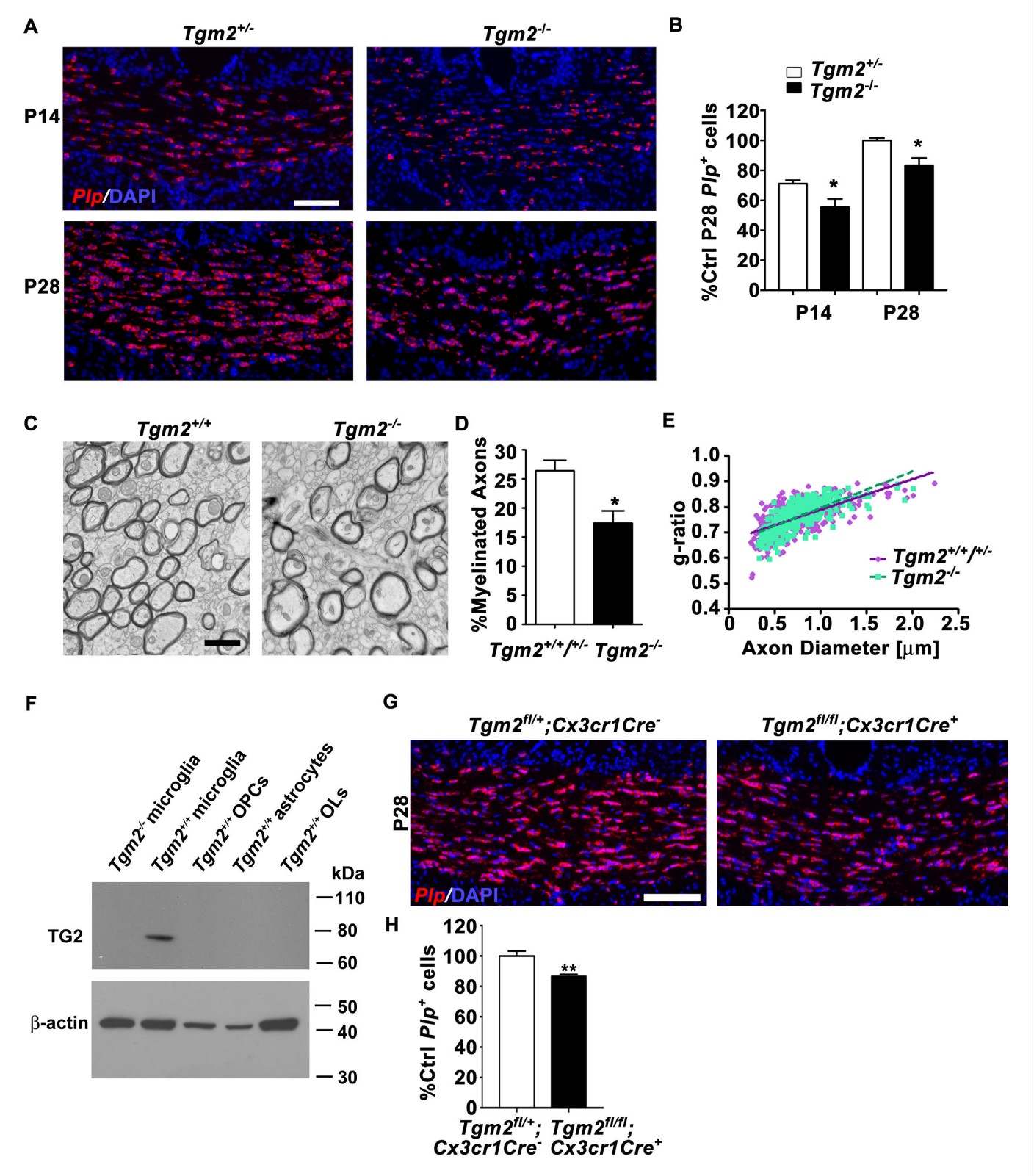

**Figure 2.** Loss of *Tgm2* leads to fewer mature OLs and hypomyelination in the CC. (A) Representative in situ hybridization (ISH) images of *Plp* (red) in the CC of P14 (top panel) and P28 (bottom panel) *Tgm2+/-* and *Tgm2-/-* mice. DAPI, blue. Scale bar, 100 μm. (B) Quantification of *Plp+* oligodendrocytes from the CC of *Tgm2+/-* and *Tgm2-/-* P14 and P28 mice. * P=0.0484; *N* = 5 per genotype (P14); *p=0.0378; *N* = 4 per genotype (P28), unpaired *t*-test. (C) Representative TEM images from P28 CC of *Tgm2+/+* and *Tgm2-/-* mice. Scale bar, 1 μm. (D) Percentage of myelinated axons in the CC of control

*Figure 2 continued on next page*

*Figure 2 continued*

(*Tgm2*[+/+] and *Tgm2*[+/-]) and *Tgm2*[-/-] mice. *p=0.0299; *N* = 3 per genotype; unpaired *t*-test. (E) Scatter plot displaying g-ratio values in the CC of control and *Tgm2*[-/-] mice. (F) TG2 western blot on acutely isolated microglia, OPCs, astrocytes, and mature oligodendrocytes from the P7 neonatal brain. β-actin was used as loading control. (G) Representative ISH images of *Plp* (red) in the CC of P28 *Tgm2*[fl/+];*Cx3cr1Cre*[-] and *Tgm2*[fl/fl];*Cx3cr1Cre*[+] mice. DAPI, blue. Scale bar, 100 μm. (H) Quantification of *Plp*[+] oligodendrocytes in the CC of P28 *Tgm2*[fl/+];*Cx3cr1Cre*[-] and *Tgm2*[fl/fl];*Cx3cr1Cre*[+] mice. **p=0.00850; *N* = 4 per genotype, unpaired *t*-test. Error bars are means ± s.e.m (B, D, H).

DOI: https://doi.org/10.7554/eLife.33385.005

The following figure supplements are available for figure 2:

**Figure supplement 1.** Oligodendrocyte number recovers by 5 month in *Tgm2* knockout mice.

DOI: https://doi.org/10.7554/eLife.33385.006

**Figure supplement 2.** Loss of *Tgm2* does not affect g-ratio, myelinated axon distribution, axon diameter or axon number.

DOI: https://doi.org/10.7554/eLife.33385.007

**Figure supplement 3.** Deleting one allele of *Tgm2* has no effect on the number of *Plp*[+] + and myelinated axons.

DOI: https://doi.org/10.7554/eLife.33385.008

**Figure supplement 4.** Developmental expression of *Tgm2*.

DOI: https://doi.org/10.7554/eLife.33385.009

proliferation was significantly increased in the presence of both rTG2 and laminin-111, but not in cells exposed to either protein alone (*Figure 3I*). Fibronectin, another ECM protein expressed in the CNS and crosslinked by TG2 (*Lorand et al., 1988*), did not enhance OPC proliferation in this assay (*Figure 3I*), suggesting specificity for laminin-111. These data support a model in which activated OPC ADGRG1 forms a signaling triad with TG2 and laminin-111 in order to support proliferation. To test this hypothesis, we generated an enzymatically dead TG2. TG2 can function as both GTPase intracellularly and crosslinking enzyme in the extracellular space (*Lorand and Graham, 2003*; *Nakaoka et al., 1994*). A single missense mutation, W241A, in the enzymatic core specifically abolishes its crosslinking enzymatic activity while preserving its GTPase function (*Pinkas et al., 2007*). Indeed, TG2-induced OPC proliferation in the presence of laminin-111 required the enzymatic crosslinking activity of TG2, as transamidation-inactive TG2-W241A mutant protein (*Pinkas et al., 2007*) failed to promote OPC proliferation (*Figure 3J*). To confirm that TG2 regulates OPC proliferation in a ADGRG1-dependent manner, we cultured OPCs derived from P5 *Adgrg1*[+/+] or *Adgrg1*[-/-] mice in the presence of laminin-111, with or without rTG2. rTG2 did not promote proliferation of *Adgrg1*[-/-] OPCs (*Figure 3K*), further supporting a ADGRG1-dependent TG2 function on OPC proliferation.

## TG2 and ADGRG1 upregulate pro-mitogenic signaling proteins

Next, we investigated intracellular mechanisms linking the interaction between ADGRG1 and TG2 with OPC cell division. Cyclin-dependent kinase 2 (CDK2) regulates OPC cell cycle progression (*Belachew et al., 2002*; *Jablonska et al., 2007*), and we previously reported reductions in CDK2 protein levels in *Adgrg1*-null OPCs (*Giera et al., 2015*). Therefore, we investigated whether loss of *Tgm2* caused similar reductions in CDK2. Indeed, western blot analysis of CDK2 in acutely isolated OPCs from P6 *Tgm2*[-/-] mice showed significantly decreased levels of CDK2 protein compared to littermate controls (*Figure 3L and M*).

An additional regulator of cell cycle progression is RhoA (*Chircop, 2014*), which was recently reported to function downstream of ADGRG1 activation in OPCs (*Ackerman et al., 2015*; *Giera et al., 2015*). Using a pull-down assay to specifically isolate activated RhoA in corpus callosal tissue of *Tgm2*[-/-] mice and littermate controls, we found that active RhoA was significantly reduced in *Tgm2*[-/-] CC (*Figure 3N and O*). This suggests that ADGRG1 activates RhoA to regulate OPC proliferation in a TG2-dependent manner.

## ADGRG1 regulates remyelination in two murine models of demyelination

Having found that the interaction of microglial TG2 with OPC-derived ADGRG1 sustained OPC proliferation during developmental myelination, we next asked if this interaction was relevant during myelin repair in the adult CNS using two established murine models of toxin-induced demyelination. The finding that *Tgm2* is highly expressed in purified adult microglia, and that this expression is not diminished in a murine model of MS additionally motivated us (*Wlodarczyk et al., 2017*) (*Figure 4—*

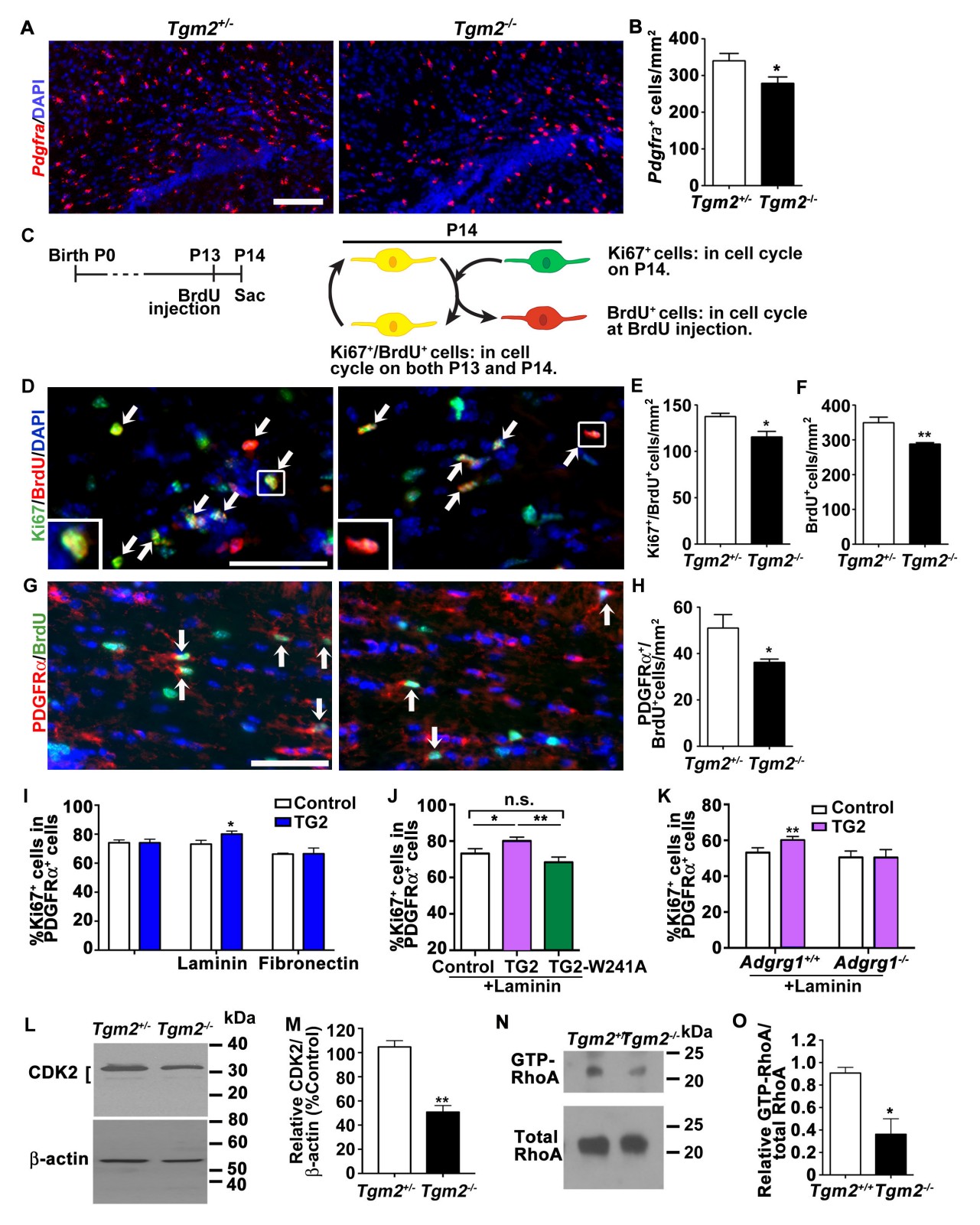

**Figure 3.** TG2 regulates OPC proliferation through ADGRG1. (**A**) Representative ISH images of *Pdgfra* (red) in the CC of P14 *Tgm2*$^{+/-}$ and *Tgm2*$^{-/-}$ mice. DAPI, blue. Scale bar, 100 μm. (**B**) Quantification of *Pdgfra*$^+$ OPCs in the CC of control and *Tgm2*$^{-/-}$ mice. *p=0.0320, unpaired t-test, *N* = 6 per genotype. (**C**) Cartoon showing the cell cycle exit assay. (**D**) Representative images of double IHC of BrdU (red) and Ki67 (green) in the CC of P14 *Tgm2*$^{+/-}$ and *Tgm2*$^{-/-}$ mice that were pulsed with BrdU 24 hr earlier (arrows mark double positive Ki67$^+$/BrdU$^+$ cells). DAPI, blue. Scale bar, 50 μm. (**E**)
*Figure 3 continued on next page*

**Figure 3 continued**

The density of Ki67$^+$/BrdU$^+$ cells in the CC is lower in $Tgm2^{-/-}$ mice. *p=0.0207, unpaired $t$-test, $N$ = 5 per genotype. (F) The density of BrdU$^+$ cells in the CC is reduced in $Tgm2^{-/-}$ mice. **p=0.0099, unpaired $t$-test, $N$ = 5 per genotype. (G) Representative images of double IHC of PDGFRα (red) and BrdU (green) in the CC of P14 $Tgm2^{+/-}$ and $Tgm2^{-/-}$ mice (arrows mark double positive PDGFRα$^+$/BrdU$^+$cells). DAPI, blue. Scale bar, 50 μm. (H) The number of BrdU$^+$/PDGFRα$^+$ cells in the CC is reduced in $Tgm2^{-/-}$ mice. *p=0.0465, unpaired $t$-test, $N$ = 4 per genotype. Error bars are means ± s.e.m. (I) The effect of recombinant TG2 (rTG2) on OPC proliferation in basal condition, laminin-111 or fibronectin. rTG2 stimulates OPC proliferation only in the presence of laminin-111. *p=0.039, paired t-test, $N$ = 3–6 per group. (J) The effect of wild type TG2 and its enzymatic dead mutant TG2-W241A proteins on OPC proliferation. Only wild type TG2 stimulates OPC proliferation. **p=0.0091; One-way ANOVA followed by Tukey post-hoc test, F(2,14) = 6.7, *p=0.05 (control vs. rTG2); n.s. p=0.70 (control vs. TG2-W241A); **p=0.0088 (TG2 vs. TG2-W241A); (K) $Adgrg1^{-/-}$ OPCs fail to respond to rTG2-enhanced proliferation. *p=0.0063, paired t-test, $N$ = 5 per genotype. (L) Western blot analyses of CDK2 in acutely isolated OPCs from P7 $Tgm2^{+/-}$ and $Tgm2^{-/-}$ brains. The bracket indicates CDK2 protein isoform bands. β-actin was used as loading control. (M) CDK2 protein levels are reduced in the $Tgm2^{-/-}$ mice. ±=0.002, unpaired t-test, $N$ = 3 per genotype. (N) Western blot of active RhoA (top panel) and total RhoA (bottom panel) in the CC of $Tgm2^{+/+}$ and $Tgm2^{-/-}$ mice. (O) The relative level of active RhoA to total RhoA was diminished in the CC of $Tgm2^{-/-}$ mice compared to $Tgm2^{+/+}$ control mice. *p=0.0207, unpaired $t$-test, $N$ = 3 per genotype. Error bars are means ± s.e.m (B, E, F, H, I, J, K, M, O).

DOI: https://doi.org/10.7554/eLife.33385.010

The following figure supplement is available for figure 3:

**Figure supplement 1.** Mutations in *gas6*, *mpp6*, and *plec* do not affect CNS Mbp expression.

DOI: https://doi.org/10.7554/eLife.33385.011

*figure supplement 1A*). Moreover, a previous report showed that *Tgm2* knockout mice exhibit impaired remyelination following cuprizone exposure (*Van Strien et al., 2011*), leading us to hypothesize that the relevant OPC receptor for TG2 might be ADGRG1 in this remyelination model.

To study whether loss of OPC *Adgrg1* impairs remyelination, we fed cuprizone to OPC-specific *Adgrg1* knockout mice and littermate controls for 6 weeks and assessed remyelination status by analyzing myelin content after 0, 3, 7, and 10 days recovery (DR). Quantitative morphometry of Black-Gold-stained myelin revealed a significant decrease in remyelination at 7 and 10 DR in the CC of mice lacking *Adgrg1* in OPCs compared with control littermates (*Figure 4A and B*). As we observed during CNS development, this reduction in myelin was accompanied by a reduction in the number of myelinating OLs (*Figure 4C* and *Figure 4—figure supplement 1B*). In the focal de-/re-myelination model using injections of lysophosphatidylcholine (LPC) into the external capsule of the CC (*Hammond et al., 2015*) to create lesions, mice lacking *Adgrg1* in OPCs had reduced numbers of myelinating OLs in the lesion 14 dpl (*Figure 4D and E*). Additionally, in the same LPC injection model, microglia-specific *Tgm2* knockout mice had fewer myelinated axons in the lesion compared to control mice (*Figure 4F and G*), while g-ratio was unaffected (*Figure 4H*).

To further evaluate whether TG2 promotes remyelination through OPC ADGRG1, we conducted studies using an ex vivo model of LPC-induced de-/re-myelination in organotypic cerebellar slices (*Birgbauer et al., 2004*), in which demyelination and repair mechanisms differ from those induced by cuprizone (*Blakemore and Franklin, 2008*). Demyelination was induced by a 24 hr exposure to LPC, and after four days recovery we observed robust remyelination in wt cerebellar slices. By comparison, this remyelination was significantly impaired in cerebellar slices derived from OPC-specific *Adgrg1*-null mice (*Figure 5A–C*). To test whether rTG2 could enhance remyelination, we added rTG2 to cerebellar slices during the recovery phase. Indeed, rTG2 increased myelination in both wt and *Tgm2* single knockout cerebellar slices with 4 days of daily treatments (*Figure 5D–F*). Excitingly, this effect appeared to be ADGRG1-dependent as increased remyelination was not observed in *Tgm2/Adgrg1* double knockout slices (*Figure 5D–F*). Taken together, the interaction between ADGRG1 and TG2 regulates efficient remyelination following cuprizone- and LPC-mediated injury.

## Discussion

Through a combined approach utilizing molecular, cellular, and developmental biology as well as unbiased proteomics, we demonstrate that microglial TG2 is the ligand of OPC-derived ADGRG1. Although the importance of OPC ADGRG1 in developmental CNS myelination has been previously demonstrated (*Ackerman et al., 2015*; *Giera et al., 2015*; *Salzman et al., 2016*), the relevant ligand remained undefined and the role of ADGRG1 in myelin repair was not addressed. Here, we demonstrate that microglia-derived TG2 provides essential ligand activity for OPC-derived ADGRG1 during

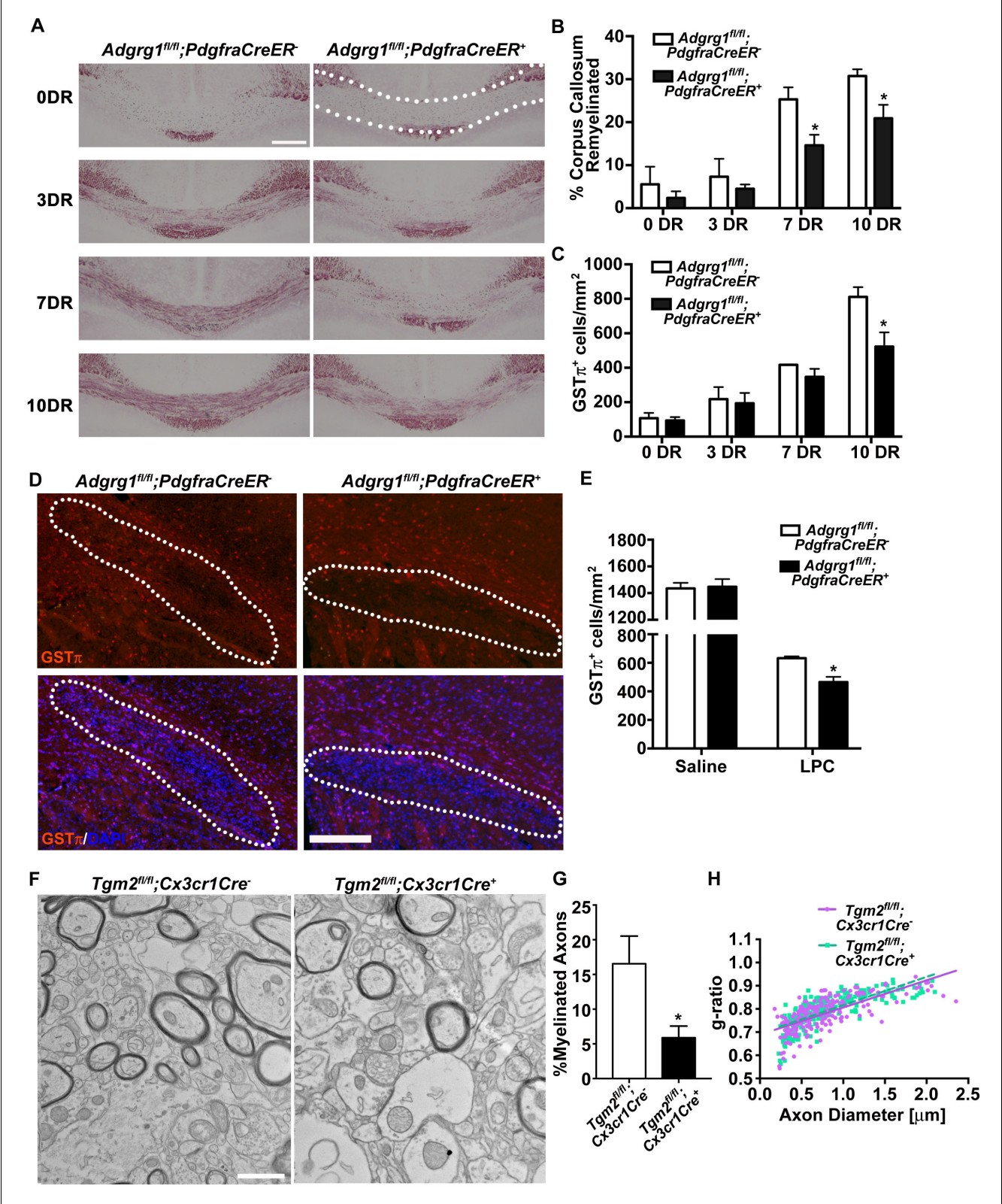

**Figure 4.** ADGRG1 is required for remyelination in vivo. (**A**) Representative images of Black-Gold myelin staining of the corpus callosum of *Adgrg1^{fl/fl}*; *PdgfraCreER^-* and *Adgrg1^{fl/fl}*;*PdgfraCreER^+* + after cuprizone feeding for 6 weeks followed by recovery for 3 d (3 DR), 7 d (7 DR) and 10 d (10 DR) (dotted line outlines quantified region). Scale bar, 250 μm. (**B**) Percentage of remyelinated corpus callosum displayed significant decrease in myelination at 7 DR and 10 DR between *Adgrg1^{fl/fl}*;*PdgfraCreER^-* and *Adgrg1^{fl/fl}*;*PdgfraCreER^+* +. *p=0.0285 (7 DR), N = 4 (cre^-); N = 4 (cre^+); *p=0.0416 (10 DR),

*Figure 4 continued on next page*

*Figure 4 continued*

$N = 3$ (cre⁻); $N = 4$ (cre⁺), unpaired *t*-test. (C) Number of GSTπ⁺ OLs are significantly decreased at 10 DR between *Adgrg1^fl/fl^;PdgfraCreER⁻* and *Adgrg1^fl/fl^;PdgfraCreER⁺* +. *p=0.0457 (10 DR), $N = 3$ (cre⁻); $N = 3$ (cre⁺), unpaired *t*-test. (D) Representative images of GSTπ⁺ OLs in the corpus callosum of *Adgrg1^fl/fl^;PdgfraCreER⁻* and *Adgrg1^fl/fl^;PdgfraCreER⁺* + 14 days post-lesion (dotted line outlines quantified region). Scale bar, 200 μm. (E) Number of GSTπ⁺ OLs are significantly decreased 14 days post-lesion between *Adgrg1^fl/fl^;PdgfraCreER⁻* and *Adgrg1^fl/fl^;PdgfraCreER⁺* +. *p=0.0133, $N = 3$ (cre⁻); $N = 3$ (cre⁺), unpaired *t*-test. (F) Representative TEM images from the CC of *Tgm2^fl/fl^;Cx3cr1Cre⁻* and *Tgm2^fl/fl^;Cx3cr1Cre* ⁺ mice. Scale bar, 1 μm. (D) Percentage of myelinated axons in the CC of *Tgm2^fl/fl^;Cx3cr1Cre⁻* and *Tgm2^fl/fl^;Cx3cr1Cre* ⁺ mice. *p=0.0493; $N = 4$ per genotype; unpaired *t*-test. (E) Scatter plot displaying g-ratio values in the CC of *Tgm2^fl/fl^;Cx3cr1Cre⁻* and *Tgm2^fl/fl^;Cx3cr1Cre* ⁺ mice.

DOI: https://doi.org/10.7554/eLife.33385.012

The following figure supplement is available for figure 4:

**Figure supplement 1.** Loss of *Adgrg1* in the OL lineage leads to reduced numbers of myelinating OLs.

DOI: https://doi.org/10.7554/eLife.33385.013

CNS myelination and that this signaling pathway is implicated in remyelination. This de-orphanization is a mandatory first step in the therapeutic exploitation of this novel pathway.

This study elucidates a novel role for microglia-ECM-OPC interactions in developmental myelination as well as in remyelination. It is known that laminin regulates oligodendrogenesis and CNS myelination by interacting with integrins and dystroglycan (*Colognato and Tzvetanova, 2011*; *Colognato et al., 2004*; *Colognato et al., 2007*; *Colognato et al., 2002*) and promotion of OPC proliferation in vivo (*Relucio et al., 2012*). Here, we present a new tripartite signaling complex – microglial TG2, ECM laminin, and ADGRG1 on OPCs – that drives OL development.

TG2 is a member of the transglutaminase protein family; family members including TG2 function as crosslinking enzymes to ligate proteins between the ε-amino group of a lysine residue and the γ-carboxamide group of a glutamine residue (*Fesus and Piacentini, 2002*; *Lai et al., 2017*). Unlike other members of the transglutaminase family, TG2 can be found both in the intracellular and the extracellular spaces of various types of tissues. The intracellular portion of TG2 is thought to predominantly act as a signaling molecule that requires its GTPase activity (*Nakaoka et al., 1994*), whereas the extracellular portion of TG2 binds to proteins of the ECM and exerts its crosslinking enzymatic activity (*Belkin, 2011*). A single missense mutation (W241A) in the catalytic core of TG2 specifically abolishes its crosslinking enzymatic activity with other functions preserved (*Pinkas et al., 2007*). Interestingly, the enzymatic dead TG2 failed to promote OPC proliferation in vitro (*Figure 3J*), supporting the model that it functions as a crosslinking enzyme in regulating OL development and CNS myelin formation and repair.

It was previously reported that TG2 binds laminin-111 (*Aeschlimann et al., 1992*). In our study, we discovered that recombinant wt TG2, but not its enzymatic-dead counterpart, stimulates OPC proliferation via ADGRG1 in the presence of laminin-111. We speculate that TG2 interactions with ADGRG1 and laminin-111 modulate ECM properties including stiffness, thus promoting OPC proliferation (*Jagielska et al., 2012*). Although our study results support that ADGRG1-TG2-laminin-111 promotes OPC proliferation, it is possible that the same signaling triad could also regulates OPC differentiation. This model will be a subject of future investigation. It is worth noting that we focused our study in the white matter. Recent reports have described differences in gene expression, morphology, and metabolism of grey and white matter microglia and OPCs (*Hagemeyer et al., 2017*; *Young et al., 2013*). It will be an interesting research direction to evaluate whether there is a difference in microglial-ECM-OPC interactions in grey and white matter.

We show that TG2, contingent on its crosslinking activity and together with laminin-111, binds to the ADGRG1 NTF and initiates G-protein signaling (*Figure 6*). Activation of ADGRG1 in OPCs activates the small GTPase RhoA, a known downstream signaling protein, and additionally promotes the accumulation of the critical pro-mitotic cell cycle regulator CDK2. The latter finding is novel and offers new avenues of investigation into aGPCR-dependent regulation of cell cycle progression, a necessary step in generating OLs during myelination or remyelination.

It is worth noting that loss of ADGRG1 does not cause as drastic OL development phenotypes compared to loss of other receptors such as GPR17 (*Chen et al., 2009*). Nevertheless, subtle fine-tuning of the developing brain is critically important for an organism as a whole: the complete loss of oligodendrocytes will be lethal, whereas dysregulation of myelin formation can cause diseases such as leukodystrophyies, and subtle dyshomeostasis could certainly disrupt brain development

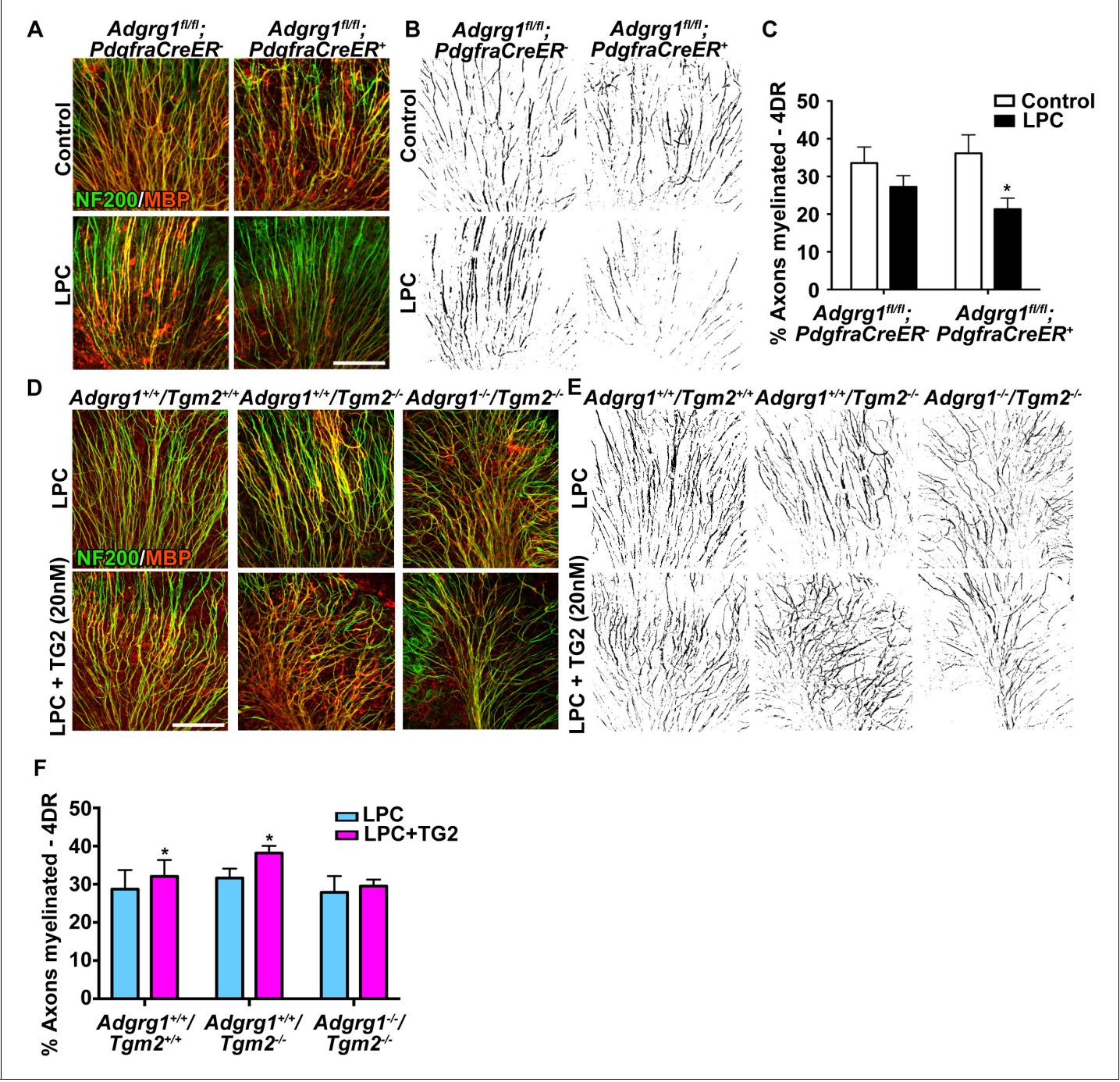

**Figure 5.** ADGRG1 and its ligand TG2 are required for remyelination. (**A**) Representative images of cerebellar slice cultures from P10 *Adgrg1^{fl/fl}*; *PdgfraCreER^-* and *Adgrg1^{fl/fl}*;*PdgfraCreER^+* + cerebella that were demyelinated with LPC for 24 hr followed by 4 days of recovery (4DR). Slices were labeled with NF200 (green) and MBP (red); myelinated fibers appear yellow in merged images. Scale bar, 100 μm. (**B**) Composite images created in Image J show myelinated axons (black) for quantification. (**C**) Percentage of myelinated axons after remyelination. Remyelination is reduced in cerebellar slices that lack OPC-derived ADGRG1. *p=0.0135, paired *t*-test, *N* = 5 per genotype. (**D**) Representative images of cerebellar slice cultures from P10 *Adgrg1^{+/+}*;*Tgm2^{+/+}*, *Adgrg1^{+/+}*;*Tgm2^{-/-}* and *Adgrg1^{-/-}*;*Tgm2^{-/-}* mouse cerebella that were demyelinated with LPC for 24 hr followed by 4 days of remyelination in the presence or absence of rTG2 (20 nM). Slices were immunostained with NF200 (green) and MBP (red); myelinated fibers appear yellow in merged images. Scale bar, 100 μm. (**E**) Composite images created in Image J show myelinated axons (black) for quantification. (**F**) Percentage of myelinated axons. Recombinant TG2 promotes remyelination in *Adgrg1^{+/+}*;*Tgm2^{+/+} and Adgrg1^{+/+}*;*Tgm2^{-/-}*, but not *Adgrg1^{-/-}*;*Tgm2^{-/-}* cerebellar slices. *p=0.0475 (*Adgrg1^{+/+}*;*Tgm2^{+/+}*), *p=0.0197 (*Adgrg1^{+/+}*;*Tgm2^{-/-}*), paired *t*-test, *N* = 5–6 per genotype. Error bars are mean ± s.e.m (**C, F**)..

DOI: https://doi.org/10.7554/eLife.33385.014

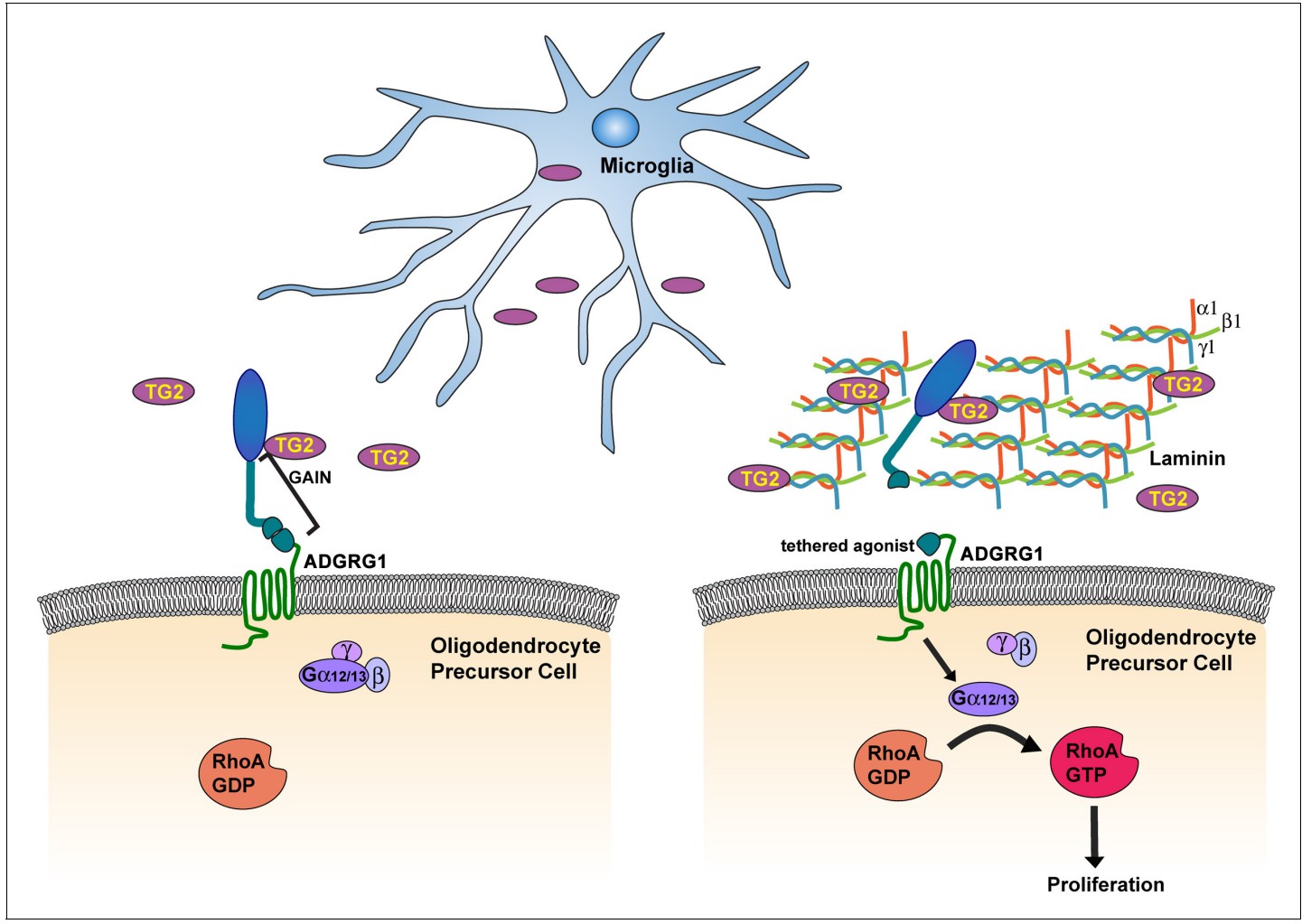

**Figure 6.** Microglia promote OPC proliferation via ADGRG1 signaling. TG2, secreted by microglia, binds ADGRG1 but fails to activate the receptor in the absence of ECM protein laminin-111. The binding of TG2 and laminin-111 to ADGRG1 leads to the dissociation of the ADGRG1 NTF from its CTF, allowing the tethered agonist to initiate G-protein signaling, culminating in activated RhoA, which promotes OPC proliferation. GAIN, GPCR-Autoproteolysis-INducing.

DOI: https://doi.org/10.7554/eLife.33385.015

(*Sharon et al., 2016*). Although perturbations to the ADGRG1-TG2 pathway had mild effects on in vitro OPC proliferation, loss-of-function mutations in *ADGRG1* manifest as CNS hypomyelination in humans (*Giera et al., 2015*; *Piao et al., 2005*; *Piao et al., 2004*), underscoring the importance of understanding ADGRG1 function in OL lineage cells.

Given the importance of myelin formation, maintenance, and repair in neurological diseases, these findings may be translated to clinical benefit in both developmental and acquired diseases of myelination. Pharmacological and biological modulators of ADGRG1 activity have recently been identified (*Stoveken et al., 2016*; *Salzman et al., 2016*). Furthermore, modulating TG2 expression and/or secretion, or developing peptidomimetic drugs could represent as viable therapeutic approaches in promoting remyelination. It will be important in future work to explore the intersection of therapeutic strategies to target aGPCRs with their cell-type-specific ligands, expression patterns, and contributions in human disease.

# Materials and methods

**Key resources table**

| Reagent type (species) or resource | Designation | Source or reference | Identifiers | Additional information |
|---|---|---|---|---|
| Strain, strain background (*M. musculus*) | *Adgrg1* knockout mice | Genentech/Lexicon Genetics | | |
| Strain, strain background (*M. musculus*) | *Tgm2* knockout mice | Dr. Gerry Melino, University of Rome, Italy PMID: 11113189 | | |
| Strain, strain background (*M. musculus*) | *Adgrg1*<sup>fl/+</sup> | Dr. Xianhia Piao, Boston Children's Hospital, Boston MA, PMID: 25607655 | | |
| Strain, strain background (*M. musculus*) | *Pdgfra*<sup>Cre/ERT</sup> | Jackson Laboratory | Cat# 018280 | |
| Strain, strain background (*M. musculus*) | *Tgm2*<sup>fl/+</sup> | Dr. Joan Cook-Mills, Northwestern University, Chicago, IL PMID: 11274171 | | |
| Strain, strain background (*M. musculus*) | *Cx3cr1*<sup>Cre</sup> | Jackson Laboratory | Cat# 025524 | |
| Antibody | mouse anti-Thy1.2 | Serotec | MCA02R | 12 µl in 10 ml |
| Antibody | mouse anti-GalC | Millipore | MAB342 | 12 µl in 10 ml |
| Antibody | mouse anti-ADGRG1 (H11) | Dr. Xianhia Piao, Boston Children's Hospital, Boston MA, PMID: 21768377 | | (1:200) (IHC) |
| Antibody | rabbit anti-ADGRG1 (199) | Dr. Xianhia Piao, Boston Children's Hospital, Boston MA, PMID: 18509043 | | (1:200) (IHC) |
| Antibody | rabbit anti-MBP | Millipore | AB980 | (1:200) (IHC) |
| Antibody | rabbit anti-Iba1 | Wako | 019–19741 | (1:400) (IHC) |
| Antibody | rabbit anti-GFAP | Abcam | ab7260 | (1:1000) (IHC) |
| Antibody | rabbit anti-NG2 | Millipore | AB5320 | (1:200) (IHC) |
| Antibody | rat anti-PDGFRα | BD Bioscience | 558774 | (1:500) (IHC) |
| Antibody | rabbit anti-PDGFRα | Cell Signaling Technologies | 3164S | (1:500) (IHC) |
| Antibody | rat anti-Ki67 | Affymetrix eBioscience | 14-5698-80 | (1:100) (IHC) |
| Antibody | rat anti-BrdU | Accurate Chemical and Scientific Corporation | OBT0030S | (1:100) (WB) |
| Antibody | mouse anti-RhoA | Cytoskeleton | ARH03-A | (1:500) (WB) |
| Antibody | mouse anti-CDK2 | Santa Cruz | sc-6248 | (1:1000) (WB) |
| Antibody | mouse anti-β-actin | Sigma | A5044 | (1:5000) (WB) |
| Antibody | mouse anti-NF200 | Sigma | N0142 | (1:500) (IHC) |
| Antibody | rat anti-MBP | Abcam | ab7349 | (1:200) (IHC) |
| Antibody | rabbit anti-GSTπ | Enzo | ADI-MSA-102-E | (1:500) (IHC) |
| Antibody | mouse anti-Ki67 | BD Bioscience | 550609 | (1:100) (IHC) |
| Antibody | goat anti mouse IgG-HRP | Sigma | A4416 | (1:3000) (WB) |
| Antibody | goat anti mouse or rabbit IgG-HRP | Sigma | A6154 | (1:3000) (WB) |
| Antibody | goat anti-mouse IgG-Alexa 488 | ThermoFisher | A-11001 | (1:1000) (IHC) |
| Antibody | goat anti-rat IgG-Alexa 488 | ThermoFisher | A-11006 | (1:1000) (IHC) |
| Antibody | goat anti-rat IgG-Alexa 546 | ThermoFisher | A-11081 | (1:1000) (IHC) |
| Antibody | goat anti-rabbit IgG-Alexa 555 | ThermoFisher | A-21428 | (1:1000) (IHC) |
| Antibody | goat anti-rabbit IgG-Alexa 546 | ThermoFisher | A-11035 | (1:1000) (IHC) |

*Continued on next page*

*Continued*

| Reagent type (species) or resource | Designation | Source or reference | Identifiers | Additional information |
|---|---|---|---|---|
| Recombinant DNA reagent | Pdgfra plasmid | Dr. Charles Stiles, Dana-Farber Cancer Institute, Boston, MA | | |
| Recombinant DNA reagent | Plp plasmid | Addgene | 22651 | |
| Sequence-based reagent | *Adgrg1* common | Integrated DNA Technologies (IDT) | 5′- CGAGAAGACTTC CGCTTCTG −3 | 20 µM |
| Sequence-based reagent | *Adgrg1* wt | Integrated DNA Technologies (IDT) | 5′- AAAGTAGCTAAG ATGCTCTCC −3′ | 20 µM |
| Sequence-based reagent | *Adgrg1* knockout | Integrated DNA Technologies (IDT) | 5′- GCAGCGCATCG CCTTCTATC −3′ | 20 µM |
| Sequence-based reagent | *Adgrg1*$^{fl/+}$ common | Integrated DNA Technologies (IDT) | 5′- TGGTAGCTAACCTACT CCAGGAGC −3′ | 20 µM |
| Sequence-based reagent | *Adgrg1*$^{fl/+}$ wt | Integrated DNA Technologies (IDT) | 5′- GGTGACTTTGGTG TTCTGCACGAC −3′ | 20 µM |
| Sequence-based reagent | *Adgrg1*$^{fl/+}$ floxed | Integrated DNA Technologies (IDT) | 5′- CACGAGACTAGTGA GACGTGCTA −3′ | 20 µM |
| Peptide, recombinant protein | laminin I | R and D Systems | 3400-010-01 | |
| Peptide, recombinant protein | fibronectin | Millipore | 341668 | |
| Commercial assay or kit | DIG RNA labeling kit | Roche Applied Science | 11175025910 | |
| Chemical compound, drug | BlackGold | Millipore | AG105 | |
| Chemical compound, drug | L-α-lyso-Lecithin, Egg Yolk | Millipore-Sigma | 440154 | LPC injections |
| Chemical compound, drug | L-α-Lysophosphatidylcholine from egg yolk | Sigma-Aldrich | L4129 | for organotypic slice culture |
| Chemical compound, drug | tamoxifen | Sigma-Aldrich | T5648 | 100 mg/kg/day tamoxifen for five consecutive days for adults; 50 mg/kg/day for three consecutive days for pups |
| Software, algorithm | Graphpad | GraphPad Software, La Jolla, CA 92037 USA | | |
| Software, algorithm | ImageJ | (http://rsb.info.nih.gov/ij/) | | |

## Mouse strains

All mice were treated according to the guidelines of the Animal Care and Use Committee at Boston Children's Hospital. The *Adgrg1* knockout mice were obtained from Genentech/Lexicon Genetics. The mutant mice were originally created in a 129/BL6 background, but were derived into the FvB strain and bred into BALB/c strain resulting in a mixed genetic background of the mutant mice of 129/BL6/FvB/BALB/c (*Li et al., 2008*). Genotyping was performed by PCR using the following primers (Integrated DNA Technologies, Coralville, IA): A (5′- CGAGAAGACTTCCGCTTCTG −3′), B (5′- AAAGTAGCTAAGATGCTCTCC −3′), and Neo (5′- GCAGCGCATCGCCTTCTATC −3′). *Tgm2* knockout mice were provided by Dr. Gerry Melino, University of Rome, Italy (*De Laurenzi and Melino, 2001*). For *Adgrg1/Tgm2* double knockout mice, *Adgrg1* knockout mice were bred with *Tgm2* knockout mice. *Adgrg1*$^{fl/+}$ mice were generated at the Mouse Gene Manipulation Core at Boston Children's Hospital (*Giera et al., 2015*). Genotyping was performed by PCR with the following primers: primer 1: 5′- TGGTAGCTAACCTACTCCAGGAGC −3′, primer 2: 5′- GGTGACTTTGG TGTTCTGCACGAC −3′ and primer 3: 5′- CACGAGACTAGTGAGACGTGCTAC −3′. *Pdgfra*$^{Cre/ERT}$ mice in a C57BL/6 background were purchased from Jackson Laboratory (Bar Harbor, ME; Cat#

018280) (*Kang et al., 2010*) and were crossed with *Adgrg1^fl/fl^* mice to generate *Adgrg11^fl/fl^*; *Pdgfra^CreERT+^* mice and their littermate controls. *Tgm2^fl/+^* mice were generously provided by Dr. Joan Cook-Mills (Northwestern University, Chicago, IL) (*Nanda et al., 2001*) and crossed with *Cx3cr1^Cre^* mice purchased from Jackson Laboratory (Bar Harbor, ME; Cat# 025524) to generate *Tgm2^fl/fl^;Cx3cr1^Cre+^* mice and their littermate controls.

## Antibodies

For IHC or western blot analyses, we used the following primary antibodies: mouse anti-ADGRG1 (H11) (1:200) (*Luo et al., 2011*) and rabbit anti-ADGRG1 (199) (1:200) (*Li et al., 2008*), rabbit anti-MBP (Millipore, Burlington, MA; Cat #AB980, 1:200), rabbit anti-Iba1 (Wako, Richmond, VA; Cat# 019–19741, 1:400), rabbit anti-GFAP (Abcam, Cambridge, MA ;Cat# ab7260, 1:1000), rabbit anti-NG2 (Millipore, Burlington, MA; Cat #AB5320, 1:200), rat anti-PDGFRα (BD Bioscience, San Jose, CA ; Cat #558774, 1:500), rabbit anti-PDGFRα (Cell Signaling Technologies, Danvers, MA; Cat #3164S, 1:500), and rat anti-Ki67 (ThermoFisher, Waltham, MA ; Cat #14-5698-80, 1:100), rat anti-BrdU (Accurate Chemical and Scientific Corporation, WESTBURY, NY; Cat #OBT0030S, 1:100), mouse anti-RhoA (Cytoskeleton, Denver, CO; Cat# ARH03-A, 1:500), mouse anti-CDK2 (Santa Cruz, Dallas, TX; Cat #sc-6248, 1:1000), mouse anti-β-actin (Sigma, St. Louis, MO; Cat #A5044, 1:5000), rabbit anti-GSTπ (Enzo Life Sciences, Farmingdale, NY; Cat#ADI-MSA-102-E, 1:500), and mouse anti-Ki67 (BD Bioscience, San Jose, CA; Cat #550609, 1:100). Secondary antibodies were goat anti-mouse or anti-rat conjugated with either Alexa 488 (ThermoFisher, Waltham, MA, 1:1000) or Alexa 546 (ThermoFisher, Waltham, MA, 1:1000) and goat anti-rabbit conjugated with Alexa 546 or 555 (ThermoFisher, Waltham, MA, 1:1000), goat anti mouse or rabbit IgG-HRP (Sigma, Cat# A4416 or A6154, 1:3000).

## OPC cultures

OPCs were isolated from mixed male and female P5-8 *Adgrg1^+/+^* or *Adgrg1^-/-^* mouse forebrains as previously described (*Watkins et al., 2008*; *Wang et al., 2001*). Briefly, OPCs were purified by negatively selecting with mouse anti-Thy1.2 (BioRad, Hercules, CA; Cat #MCA02R) and mouse anti-GalC (Millipore, Burlington, MA; Cat #MAB342), followed by mouse anti-O4 (O4 hybridoma supernatant) for positive selection. After releasing OPCs from the O4 plate by trypsinization, cells were resuspended in proliferation media containing PDGF-AA and NT-3 (PreproTech, Rocky Hill, NJ ). OPCs were plated on coverslips coated with poly-D-lysine before coating with laminin I (R and D Systems, Minneapolis, MN; Cat#3400-010-01) or fibronectin (Millipore, Burlington, MA; Cat#341668) as previously described (*Dugas et al., 2006*). After 24 hr, rTG2 (2 nM) was added to the cultures and incubated for an additional 24 hr before fixation with 4% PFA and staining for PDGFRα and Ki67.

## Mixed glia culture

Mixed glia culture was isolated from P1 male and female wt forebrains, as previously described (*Giera et al., 2015*; *O'Meara et al., 2011*). Briefly, forebrains were dissociated and resulting cell suspension was cultured for 10 days in DMEM containing 10% FBS, 1% GlutaMax at 37°C and 8.5% $CO_2$. The culture was shaken for 20 hr at 250 rpm and 37°C to remove OPCs. Mixed glial cells were cultured for another 5–7 days before being washed twice with PBS and collected.

## Microglia isolation

Microglia were isolated as previously described (*Cardona et al., 2006*). Briefly, male and female P7 pups were perfused with ice-cold HBSS, before isolating and mincing the brains. The tissue was homogenized with a dounce homogenizer containing RPMI media before being mixed with stock isotonic percoll (SIP) creating a 30% SIP solution. This cell/SIP mixture was layered on top of a 70% SIP solution before being centrifuged. The resulting interphase containing the microglia was collected and washed with HBSS and cells were counted before being lysed for western blot analysis.

## Cuprizone treatment

*Adgrg1^fl/fl^;Pdgfra^CreERT-^* and *Adgrg1^fl/fl^;Pdgfra^CreERT+^* adult mice (8–10 weeks old) were injected with 100 mg/kg/day tamoxifen for five consecutive days. Mice were fed a diet containing 0.2% (w/w) cuprizone (Envigo, Indianapolis, IN) ad libitum for 6 weeks. A second pulse of tamoxifen was given

at 4 weeks of cuprizone exposure, for 3 days with 100 mg/kg/day. After 6 weeks, mice were returned to normal feed for 3, 7 or 10 days of recovery (DR). Mice were perfused as described above at 0, 3, 7, 10 DR.

## Black-Gold myelin staining

Coronal brain sections were stained with Black Gold myelin stain (Millipore, Burlington, MA; Cat# AG105) according to the manufacturer's protocol. Sections were imaged using Nikon Eclipse 80$i$ microscope (Nikon, Melville, NY). Representative images were obtained with the same exposure setting for control and mutant. Images of the corpus callosum were analyzed using NIH Image J software. The percentage of remyelinated area of the total corpus callosum was determined using a threshold procedure.

## LPC injections

$Adgrg1^{fl/fl};Pdgfra^{CreERT-}$ and $Adgrg1^{fl/fl};Pdgfra^{CreERT+}$ male mice (8–10 weeks old) were injected with 100 mg/kg/day tamoxifen for five consecutive days from −6 dpl to −1 dpl. Mice ($Adgrg1^{fl/fl};Pdgfra^{CreERT-}$ and $Adgrg1^{fl/fl};Pdgfra^{CreERT+}$ or $Tgm2^{fl/fl};Cx3cr1^{Cre-}$ and $Tgm2^{fl/fl};Cx3cr1^{Cre+}$,8–10 weeks old) were anesthetized using isoflurane (3%) before either lysolecithin (LPC, 1% in saline, 2 µl, Millipore, Burlington, MA) or saline was injected bilaterally using a Hamilton syringe in a stereotaxic apparatus at the following coordinates 1 mm anterior to bregma, 1.5 mm posterior to bregma and 3.0 mm deep as previously described (*Hammond et al., 2015*). Day of injection was denoted as 0 dpl. Mice were left for 14 dpl and subsequently perfused for either immunohistological analysis or transmission electron microscopy.

## Cerebellum slice culture and LPC treatment

$Adgrg1^{fl/fl};Pdgfra^{CreERT-}$ and $Adgrg1^{fl/fl};Pdgfra^{CreERT+}$ P7 pups were injected with tamoxifen (50 mg kg$^{-1}$) for 3 days before the cerebellar harvesting. All cerebella were harvested on P10 and placed in ice-cold Hank's buffer (ThermoFisher, Waltham, MA) before being cut into 300 µm thick sagittal sections with a vibratome (Leica Biosystems, Buffalo Grove, IL). Three slices were placed onto inserts (Millicell 0.4 µm, Millipore, Burlington, MA) in media containing 50% MEM, 25% Horse Serum, 25% Hank's Buffer, 1% GlutaMax, 5 mg/ml Glucose. After 48 hr, the slices were incubated with 0.25 mg/ml lysolecithin (Sigma-Aldrich, St. Louis, MO ) for 24 hr. Afterward the slices were washed with media and remyelination was allowed to occur for 4 days, with or without 20 nM rTG2 protein as indicated in the figures by adding rTG2 to the media and slices. Slices were fixed with 4% PFA for 20 min, permeablized with 10% Triton-X 100 for 20 min and blocked with 10% goat serum, 1% bovine serum albumin (BSA) and 0.1% Triton X-100 in PBS for 1 hr at room temperature before being incubated with the primary antibody for neurofilament (mouse anti-NF200 antibody, SigmaAldrich St. Louis, MO ; Cat#N0142 1:500) and myelin (rat anti-MBP, Abcam, Cambridge, MA; Cat#ab7349, 1:200) overnight at 4°C. Primary antibodies were visualized by incubating the sections with the appropriate fluorophore-conjugated secondary antibody for 1 hr at room temperature followed by staining of the nuclei with Hoechst 33342 (1:2000, ThermoFisher, Waltham, MA). After mounting the slices, images were taken with Zeiss LSM 700 confocal microscope. Quantification of myelinated fibers was performed using NIH Image J by subtracting background and applying threshold for both red and green channel, before measuring both myelinated and unmyelinated axon fibers.

## Histology analyses

Mouse brains were harvested after perfusion, post-fixed with 4% PFA overnight, cryoprotected with 30% sucrose, and embedded in OCT. IHC was carried out as previously described (*Jeong et al., 2012*). Briefly, after antigen retrieval in Retrievagen A Solution (BD Bioscience, San Jose, CA ), brain sections were washed with 1x PBS, blocked with 10% goat serum, 1% bovine serum albumin (BSA) and 0.1% Triton X-100 in PBS for 1 hr at room temperature before being incubated with the primary antibody overnight at 4°C. Primary antibodies were visualized by incubating the sections with the appropriate fluorophore-conjugated secondary antibody for 1 hr at room temperature followed by staining of the nuclei with Hoechst 33342 (1:2000, ThermoFisher, Waltham, MA ).

In situ hybridization was performed on 12 µm brain sections as previously described (*Dugas et al., 2010*; *Bialas and Stevens, 2013*). Probes targeting *Plp* (Addgene, Cambridge, MA; Cat #22651) and *Pdgfra* (kind gift from Charles Stiles) were generated by digesting plasmids with EcoRI and HindIII, respectively. Sp6 and T7 polymerase were used to generate DIG-labeled RNA probes in vitro transcription (Roche Applied Science, Indianapolis, IN; DIG RNA labeling kit) as per manufacturer's instructions. Hybridization occurred at 68 °C and washes at 65 °C. To detect the DIG-labeled probes, the TSA-Plus Cyanine three labeling system (Perkin Elmer, Waltham, MA) was used according to the manufacturer's instructions.

Ligand binding in situ was performed on male and female P5 wt brains that were embedded into low-melting agarose and sectioned into 500 µm thick sections using a vibratome. Sections were washed with PBS before blocking for 30 min with 0.5% goat serum and 0.1% BSA, incubating with ADGRG1$^N$-hFc for 30 min, followed with the appropriate secondary antibody for 30 min. Sections were fixed with 4% PFA for 30 min before being blocked with 10% goat serum, 1% BSA and 0.1% Triton-X 100 for 1 hr. Tissue was incubated with a marker for co-staining overnight at 4°C followed with the appropriate secondary antibody for 30 min.

All images were captured using a confocal LSM 510 NLO system or a Nikon Eclipse T*i* inverted microscope (Nikon, Melville, NY). Representative images were obtained with the same exposure setting for control and mutant.

## Cell cycle exit assay

Cell cycle exit assays were performed as previously described (*Giera et al., 2015*). Briefly, proliferating cells were labeled with BrdU (50 mg kg$^{-1}$) by intraperitoneal injection of male and female P13 *Tgm2$^{-/-}$* pups and their littermate controls. Mouse brains were harvested 24 hr later, perfused, postfixed with 4% PFA, cryoprotected with 30% sucrose, and embedded in OCT. Brain sections were processed for IHC with anti-BrdU and anti-Ki67 antibodies.

## Western blot and GTP-Rho Pull-Down assay

The corpus callosa were dissected under a Leica stereo microscope (MZ 6; Leica Biosystems, Buffalo Grove, IL), followed by washes in 1x PBS and lysis in ice-cold RIPA buffer (1% Nonidet P-40, 50 mM Tris pH 7.6, 120 mM NaCl, 1 mM EDTA) containing protease inhibitor cocktail set 1 (Millipore, Burlington, MA). The lysates were cleared of insoluble materials by centrifugation at 16,000 x *g* for 10 min at 4°C. Protein concentration was determined by a Bio-Rad protein assay method (Bio-Rad, Hercules, CA) according to the manufacturer's protocol, and equal amounts of protein were used for SDS–PAGE and western blot analysis. The GTP-Rho pull-down assay was performed as previously described (*Luo et al., 2011*); in short, tissues were pulverized on liquid nitrogen, lysed in 300 µl of ice-cold RIPA buffer containing protease inhibitors with a cell disruptor for 10 min and homogenized with a 26 G syringe needle. Equal amounts of total protein were incubated with 60 µg GST-RBD beads (Cytoskeleton, Denver, CO) at 4°C for 90 min. The beads were washed twice with lysis buffer and once with TBS buffer. Bound Rho proteins were eluted by Laemmli sample buffer and detected by western blot using mouse monoclonal anti-RhoA antibody (Cytoskeleton, Denver, CO).

## Purification of ADGRG1 immunocomplexes

ADGRG1$^N$ and ADGRG1$^{Ndel}$ fusions proteins with either mFc or hFc tag were generated as previously described (*Luo et al., 2011*). BirA enzyme was used to in vitro biotinylate ADGRG1$^N$-mFc-biotag protein as previously described (*Stacey et al., 2002*). Mixed glial cell lysate was incubated with biotinylated ADGRG1$^N$ protein in lysis buffer (20 mM Hepes [pH 7.3], 150 mM NaCl, and 5 mM MgCl$_2$) with protein inhibitor mixture (Roche Applied Science, Indianapolis, IN) and 1% Brij 96 (Sigma, St. Louis, MO). Streptavidin beads (Sigma, St. Louis, MO) were used to affinity purify immunocomplexes. ADGRG1-associated proteins were eluted in 2x SDS loading buffer, subjected to SDS/PAGE. The whole lane was used for MS at the Taplin Biological Mass Spectrometry at Havard Medical School. Mass spectrometry data were searched against the mouse International Protein Index (IPI mouse 339) database using the protein identification software Mascot (v2.2.04, Matrix Science, Boston, MA)(*Luo et al., 2011*).

## Zebrafish stocks and rearing conditions

Zebrafish (*Danio rerio*) were maintained in the Washington University Zebrafish Consortium facility (http://zebrafish.wustl.edu/). All experiments were performed in compliance with Washington University's institutional animal protocols. Embryos were collected from pair-wise or harem matings and reared at 28.5°C in egg water (5 mM NaCl, 0.17 mM KCl, 0.33 mM CaCl$_2$, 0.33 mM MgSO$_4$). All zebrafish lines used in these studies were generated in the wt (AB) background, including: *gas6*[stl228], *mpp6a*[stl233], *mpp6b*[stl234], *pleca*[stl261], and *plecb*[stl236].

## Zebrafish mutant generation

Zebrafish guide RNAs (gRNAs) were designed and generated and *cas9* RNA was synthesized at the Genome Engineering and iPSC center at Washington University School of Medicine (St. Louis, MO). gRNA sequences were as follows:

*gas6*: 5'- CAGAACCCGCAGAGCCAACCAGG −3'
*mpp6a*: 5'- GGCGGTTCAAAGTAATAACGTGG −3'
*mpp6b*: 5'- GCAGCAGGTGTTGGATAACC −3'
*pleca*: 5'- AAAACTAGGGAATAAGACTG −3'
*plecb*: 5'- TCCCTCCTGGAGGTCCTCTC −3'

Synthetic CRISPR guide mRNA was combined with *cas9* synthetic mRNA and injected into 1 cell stage wt embryos (AB) at 100 pg quantities. To recover germ-line transmitted mutations, injected founders (F0s) were grown to adulthood, outcrossed to wt AB partners, and genomic DNA was extracted from individual F1 embryos for PCR amplification and restriction digest analysis of the targeted region. Genomic DNA from F1 embryos that showed disruption of the target site was then cloned using the TOPO TA Cloning kit (Invitrogen) and Sanger sequenced.

## Zebrafish genotyping

To genotype individual larvae for phenotypic analyses, we used the following primers to amplify a 505 bp fragment of *gas6* from genomic DNA: 5'- CAGAAGAGCGAAAGTTTGAC −3' and 5'- CACAGTGAACATCATCGAGT - 3'. For *gas6*[stl228/stl228], we performed restriction digest analysis using SexAI, which cleaves wt into a 219 bp and a 286 bp fragment, but is unable to cut mutant (501 bp). To amplify *mpp6a*, we used the following primers to amplify a 317 bp fragment from genomic DNA: 5'- ACCCAGAGGCACTTGATTA −3' and 5'- GGTTCCTTCAGGATGTTAGA - 3'. For *mpp6a*[stl233/stl233], we performed restriction digest analysis using HpyCH4IV, which cleaves wt into a 93 bp and a 224 bp fragment, but is unable to cut mutant (316 bp). To amplify *mpp6b*, we used the following primers to amplify a 400 bp fragment from genomic DNA: 5'- GGAAATGACCTCAGCAGAT −3' and 5'- CATGCGTTTACCTCATTAGC - 3'. For *mpp6b*[stl234/stl234], we performed restriction digest analysis using BstNI, which cleaves wt into a 153 bp and a 247 bp fragment, but is unable to cut mutant (394 bp). To amplify *pleca*, we used the following primers to amplify a 660 bp fragment from genomic DNA: 5'- GTGGCCTTACATGACATCTT −3' and 5'- CTGAATGCTCACACAATCAC - 3'. For *pleca*[stl261/stl261], we performed restriction digest analysis using DdeI, which cleaves wt into a 248 bp and a 412 bp fragment, but is unable to cut mutant (640 bp). To amplify *plecb*, we used the following primers to amplify a 182 bp fragment from genomic DNA: 5'- ACTCATGGACTAACGTCTGC −3' and 5'- GTTTTGACGTGGGATTAGAG - 3'. For *plecb*[stl236/stl236], we performed restriction digest analysis using Hpy188III, which cleaves wt into a 66 bp and a 103 bp fragment, but is unable to cut mutant (169 bp).

## Whole mount in situ hybridization

Whole-mount ISH was performed using standard protocols (*Thisse and Thisse, 2008*). In brief, embryos were fixed at 5 dpf in 4% PFA at 4° C overnight, and then washed into 100% methanol for dehydration. Following dehydration, embryos were washed in 0.2% PBS-Tween (PBSTw), permeabilized in proteinase K (20 mg/μl diluted 1:1000 in 0.2% PBSTw), and incubated with a Digoxygenin-labeled riboprobe again *mbp* (*Lyons et al., 2005*) overnight at 65° C in hybridization buffer (50% formamide). Following probe treatment, embryos were washed to remove formamide, blocked in 2% blocking medium supplemented with 10% normal sheep serum and 0.2% Triton, and incubated in primary antibody (anti-DIG, Fab fragments (1:2000), Product # 11214667001, Roche Applied Science, Indianapolis, IN) overnight in block. Following primary antibody treatment, embryos were

washed in maleic acid buffer with 0.2% triton, and developed by alkaline phosphatase treatment. After complete, embryos were post-fixed in 4% PFA and stored long-term in 70% glycerol. Embryos were mounted on slides and imaged at 10x with an AxioCam MRm on a light microscope (Zeiss Axio Imager M2, Zeiss, San Diego, CA). Genotypes were obtained subsequent to larval analyses.

## Transmission electron microscopy

For mice, postnatal male brains were fixed by perfusion followed by immersion in a mixture of 2% glutaraldehyde and 4% paraformaldehyde in 0.1M sodium cacodylate buffer, pH 7.4. After overnight fixation, brains were postfixed with 1% osmium tetroxide/1.5% potassium ferrocyanide for 1 hr, and after washing samples, were incubated in 1% aqueous uranyl acetate for 1 hr followed by subsequent dehydration in alcohol. The samples were incubated in propylene oxide for 1 hr and infiltrated overnight in a 1:1 mixture of propylene oxide and TAAB Epon (Marivac Canada Inc., St. Laurent, Canada). The next day, samples were embedded in TAAB Epon and polymerized at 60°C for 48 hr. Ultrathin sections (~60 nm) were cut on a Reichert Ultracut-S microtome, mounted on copper grids, stained with lead citrate, and examined with a JEOL 1200EX transmission electron microscope. Images were recorded with an AMT 2 k CCD camera. The photographs were analyzed using NIH Image J Software (http://rsb.info.nih.gov/ij/) to calculate g-ratio and axon diameter. G-ratio was calculated as previously described (*Roy et al., 2007*).

## Statistical analysis

For all studies, images were scored blinded to genotype and treatments prior to quantifications. For mouse studies, data is represented as mean ± s.e.m and asterisks indicate significance: ***$p \leq 0.001$; **$p \leq 0.01$; *$p \leq 0.05$. GRAPHPAD Prism Software (GraphPad Software, La Jolla, CA) was used to determine statistical significance between genotypes and treatments using unpaired or paired Student's *t*-tests, two-tailed and unequal variance depending on animals either being paired prior to data collection or not. For in vitro culture, animals were paired prior to isolation of OPCs. One-way ANOVA followed by Tukey post-hoc was used to analyze multiple treatment condition experiments. Sample size was not pre-determined by statistical methods, but was based on similar studies in the field.

## Acknowledgements

This research was supported in part by NINDS grants R56 NS085201 (XP), R01 NS085201 (XP); National Multiple Sclerosis Society (NMSS) RG-1501–02577 (XP); NMSS Postdoctoral fellowship Award FG 2063-A1/2 (SG); NINDS F31 NS087801 (SDA); NINDS grant R01 NS079445 (KRM); and the NMSS Harry Weaver Neuroscience Fellowship (KRM); KBRI Basic Research program funded by Ministry of Science and ICT 18-BR-02-02 (SJJ); National Research Foundation of Korea 2015M3C7A1029037 (SJJ). We thank Drs. Jean-Christophe Delpech and Allison R Bialas for their critical reading of the manuscript and Timothy R. Hammond for his technical assistance with the focal demyelination mouse model.

## Additional information

### Competing interests

Beth Stevens: Reviewing editor, *eLife*. The other authors declare that no competing interests exist.

### Funding

| Funder | Grant reference number | Author |
| --- | --- | --- |
| National Institute of Neurological Disorders and Stroke | NS085201 | Xianhua Piao |
| National Institute of Neurological Disorders and Stroke | NS085201 (R56) | Xianhua Piao |
| National Multiple Sclerosis Society | FG 2063-A1/2 | Stefanie Giera |

| National Institute of Neurological Disorders and Stroke | NS087801 | Sarah D Ackerman |
| National Research Foundation of Korea | 2015M3C7A1029037 | Sung-Jin Jeong |
| Ministry of Science, ICT and Future Planning | KBRI Basic Research program 18-BR-02-02 | Sung-Jin Jeong |
| National Institute of Neurological Disorders and Stroke | NS079445 | Kelly R Monk |
| National Multiple Sclerosis Society | Harry Weaver Neuroscience Fellowship | Kelly R Monk |
| National Multiple Sclerosis Society | RG-1501-02577 | Xianhua Piao |

The funders had no role in study design, data collection and interpretation, or the decision to submit the work for publication.

### Author contributions

Stefanie Giera, Data curation, Formal analysis, Funding acquisition, Visualization, Writing—original draft, Project administration, Writing—review and editing; Rong Luo, Sarah D Ackerman, Formal analysis, Investigation, Methodology, Writing—review and editing; Yanqin Ying, Formal analysis, Investigation, Methodology; Sung-Jin Jeong, Data curation, Methodology; Hannah M Stoveken, Beth Stevens, Resources, Methodology; Christopher J Folts, Writing—review and editing; Christina A Welsh, Methodology, Writing—review and editing; Gregory G Tall, Resources, Methodology, Writing—review and editing; Kelly R Monk, Resources, Funding acquisition, Investigation, Methodology, Writing—review and editing; Xianhua Piao, Conceptualization, Funding acquisition, Writing—original draft, Project administration, Writing—review and editing

### Author ORCIDs

Stefanie Giera http://orcid.org/0000-0002-7414-0562
Sarah D Ackerman http://orcid.org/0000-0001-8752-8972
Christopher J Folts http://orcid.org/0000-0002-0448-3711
Christina A Welsh https://orcid.org/0000-0001-9802-725X
Beth Stevens http://orcid.org/0000-0003-4226-1201
Xianhua Piao http://orcid.org/0000-0001-7540-6767

### Ethics

Animal experimentation: This study was performed in accordance to the guidelines of the Animal Care and Use Committee (IACUC) protocols (17-12-3578R and 17-03-3378R) at Boston Children's Hospital. Zebrafish experiments were performed in compliance with Washington University's Institutional Animal Care and Use Committee (IACUC) protocol (20160174)

### Decision letter and Author response

Decision letter https://doi.org/10.7554/eLife.33385.018
Author response https://doi.org/10.7554/eLife.33385.019

## Additional files

### Supplementary files

• Transparent reporting form
DOI: https://doi.org/10.7554/eLife.33385.016

### Data availability

All data generated or analysed during this study are included in the manuscript and supporting files.

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
