## [Decision Letter]

Thank you for submitting your article "Microglial transglutaminase-2 drives myelination and myelin repair via GPR56/ADGRG1 in oligodendrocyte precursor cells" for consideration by *eLife*. Your article has been favorably evaluated by Eve Marder (Senior Editor) and three reviewers, one of whom is a member of our Board of Reviewing Editors. The reviewers have opted to remain anonymous.

The reviewers have discussed the reviews with one another and the Reviewing Editor has drafted this decision to help you prepare a revised submission.

The manuscript from Giera and colleagues describes a novel signaling pathway between microglia and oligodendrocyte lineage cells that promotes CNS myelination and remyelination. Previously, this group identified an important cell autonomous role for the adhesion G protein-coupled receptor GPR56 in oligodendrocyte development, but the relevant ligand for this receptor was unknown. In the current study, the investigators have identified one GPR56 ligand that appears to be critical and which points to a role of microglial cells in myelination control. Using the N-terminal fragment of recombinant GPR56 as a probe, the authors could label by IHC microglia in the developing white matter. By mass-spectrometry they identified as one GRP56 binding partner the protein TG2 (the product of the *Tgm2* gene), a transglutaminase known to interact with GPR56 on melanoma cells. They studied *Tgm2* KO mice, which are as hypomyelinated as GPR56 KO mice, with slightly fewer (proliferating) OPC, also as previously shown for GPR56 KO mice. By CRISPR mutagenesis in zebrafish, the authors then sorted out other candidates from their interaction screen.

Interestingly, recombinant TG2 stimulated OPC proliferation in vitro only in the presence of laminin and the authors suggest that laminin is part of a tripartite signaling complex. The downstream signal seems to involve CDK2 and RhoA activity. Finally, the GPR56 KO mouse shows reduced recovery (myelin repair) from cuprizone and other forms of adult demyelination.

Overall, the study is well done and will be of considerable interest. The identification of molecular insight into the influence of microglia on the CNS myelination process is of particular significance. Addressing the following points would make the manuscript more complete.

Major points:

1) The authors test the role of TG2 in the myelination/remyelination processes using mouse models, and they use zebrafish mutants to test the remaining GPR56 candidate ligands (Table 1), none of which displayed an abnormal myelin phenotype. It would have been more thorough to include a TG2 mutant also in their zebrafish screen, which should phenocopy the GPR56 mutants and provide a positive control.

2) The impression was shared by the referees that the phenotype of GPR56 and *Tgm2* mutant mice is similar but not identical, which is an important issue if one wants to understand the role of GPR56 in the corresponding human leukodystrophy. Is the *Tgm2* KO phenotype an arrest of development or a significant delay (same in the Cuprizone model)?

3) The authors assess remyelination in their cuprizone studies using Black-Gold staining, which provides an indication that the TG2 mutants remyelinate less efficiently. Nevertheless, an EM analysis of these animals would greatly strengthen the conclusion. This is particularly important since the effects of the TG2 mutation on developmental myelination is relatively mild.

4) The evidence for a "signaling triad" is suggestive, but not yet conclusive. If the functional "transglutaminase" domain of TG2 is mutated, this will likely delete the entire function of TG2. Why does that prove a direct interaction with laminin? *Tgm2* can theoretically have another important target protein, and laminin could have a distinct role in driving OPC differentiation. That should be discussed.

5) The authors demonstrate that recombinant TG2 added exogenously was able to enhance remyelination in the TG2 mutant slice culture experiments. Does a similar approach enhance remyelination of WT cultures? This would support a possible therapeutic benefit of enhancing GPR56 signaling. Are the microglial-ECM-OPC interactions the same in gray and white matter?

6) It is a little unfortunate that we have learned little about the role of TG2 and microglia in this aspect of myelination control. Is the expression of *Tgm2* developmentally regulated in microglia and is its expression affected by the demyelinating injury?

---

## [Author Response]

Major points:1) The authors test the role of TG2 in the myelination/remyelination processes using mouse models, and they use zebrafish mutants to test the remaining GPR56 candidate ligands (Table 1), none of which displayed an abnormal myelin phenotype. It would have been more thorough to include a TG2 mutant also in their zebrafish screen, which should phenocopy the GPR56 mutants and provide a positive control.

The research with mice and zebrafish were carried out simultaneously in labs of Piao and Monk, respectively. We made the conscious decision at the start of the research to not make *tgm2* mutant zebrafish based on the following rationales: (1) TG2 is a known binding partner of GPR56 in melanoma; (2) there was a published report (van Strien et al., 2011) documenting the role of TG2 in CNS remyelination, but with unknown molecular mechanism; (3) Piao’s lab has already imported both constitutive and floxed *Tgm2* mice.

We were not able to make the *tgm2* fish mutant in a timely manner for this revision due to the fact that: (1) it will take about 1 year to generate homozygous mutant fish and complete the phenotype analysis in the best case scenario and (2) Monk’s lab has recently relocated to Vollum Institute, OHSU, and it would take substantially longer time for the work to be completed given ongoing construction of the lab’s zebrafish facility. Given that mouse is a mammalian organism with closer relevance to human disease, we feel that our observations made in mice fully support our conclusion and carries significance in potential translational research, even in the absence of the supplemental data from zebrafish mutants.

2) The impression was shared by the referees that the phenotype of GPR56 and Tgm2 mutant mice is similar but not identical, which is an important issue if one wants to understand the role of GPR56 in the corresponding human leukodystrophy. Is the Tgm2 KO phenotype an arrest of development or a significant delay (same in the Cuprizone model)?

The reviewers raise a valid point. We have addressed their comments in two ways.

First, we analyzed the density of mature oligodendrocytes by in situhybridization as labeled by *Plp* in the corpus callosum of aged *Tgm2* mice (4-5 months). We found that, similar to our observations with *Gpr56* knockout mice (Giera et al., 2015), there is no significant difference in *Plp^+^*cell density between *Tgm2* knockout and wild-type mice at this age (Figure 2—figure supplement 1, *p* = 0.807). Thus, loss of *Tgm2* phenocopies the delayed maturation phenotype we observe in *Gpr56* knockouts. Second, van Strien and colleagues (van Strien et al., 2011) report that no significant difference exists in the density of *Plp*^+^ oligodendrocytes in two-month old mice prior to exposure to a cuprizone diet, whereas *Tgm2* knockout mice showed impaired remyelination in the cuprizone model. We have better incorporated these findings into our manuscript.

3) The authors assess remyelination in their cuprizone studies using Black-Gold staining, which provides an indication that the TG2 mutants remyelinate less efficiently. Nevertheless, an EM analysis of these animals would greatly strengthen the conclusion. This is particularly important since the effects of the TG2 mutation on developmental myelination is relatively mild.

We agree with the reviewers’ request and have since performed additional experiments. We note that the brains collected from our cuprizone experiments were fixed for routine histology study and are therefore not suitable for EM analysis. Given that collecting a new cohort of cuprizone animals would take a significant amount of time, we instead incorporated the lysolecithin model of focal de-/re-myelination into our manuscript to address this point. This additional study further strengthened our conclusion by (1) confirming the role of GPR56-TG2 in CNS myelin repair using a different in vivo de-/re-myelination model and (2) providing both histology and EM data documenting impaired remyelination in the absence of either OPC *Gpr56* or microglial *Tgm2* in a different myelin repair model. The new data is presented in Figure 4D-H.

4) The evidence for a "signaling triad" is suggestive, but not yet conclusive. If the functional "transglutaminase" domain of TG2 is mutated, this will likely delete the entire function of TG2. Why does that prove a direct interaction with laminin? Tgm2 can theoretically have another important target protein, and laminin could have a distinct role in driving OPC differentiation. That should be discussed.

We thank reviewers for this suggestion. TG2 is a multifaceted molecule with diverse functions. For example, it functions as a GTPase intracellularly and as a crosslinking enzyme in the extracellular space. A single missense mutation – W241A – in the catalytic core specifically abolishes its crosslinking enzymatic activity while preserving its GTPase function (Lorand and Graham, 2003; Nakaoka et al., 1994; Pinkas et al., 2007). In this revised manuscript, we included more detailed description of the enzymatic dead TG2 mutant (W241A) in both the Results and Discussion sections to clarify this concept.

5) The authors demonstrate that recombinant TG2 added exogenously was able to enhance remyelination in the TG2 mutant slice culture experiments. Does a similar approach enhance remyelination of WT cultures? This would support a possible therapeutic benefit of enhancing GPR56 signaling. Are the microglial-ECM-OPC interactions the same in gray and white matter?

We have performed the requested experiment and found that the addition of recombinant TG2 to wild-type demyelinated slices provides a modest increase in remyelination compared to vehicle-treated controls; we have included these data in Figure 5D-F. While our findings raise the theoretical possibility of using TG2 therapeutically, it is worth noting that although the exogenous addition of recombinant TG2 in our organotypic slice assays is suggestive of its therapeutic potential, it is a conceivable challenge in regards to tissue penetrance or degradation of TG2 protein in vivo in a whole organism. Alternative approaches to enhance TG2 expression and/or secretion, or peptidomimetic drugs, might be worthwhile avenues of future investigation.

We appreciate the reviewers’ comments on the differences between microglial-ECM-OPC interactions in grey and white matter. Indeed, recent reports have described differences in gene expression, morphology, and metabolism of grey and white matter microglia and OPCs (Young et al., 2013; Hagemeyer et al., 2017), and the ECM composition differs between myelinated and unmyelinated regions of the CNS. While we consider this an interesting consideration, we believe its focus lies outside the scope of this manuscript. Therefore, we only add discussion of this possibility and appropriate citations in the Discussion section.

6) It is a little unfortunate that we have learned little about the role of TG2 and microglia in this aspect of myelination control. Is the expression of Tgm2 developmentally regulated in microglia and is its expression affected by the demyelinating injury?

The reviewers raise an important consideration. To address this, we mined recently published RNA-sequencing databases for the microglial expression of *Tgm2* (Zhang et al., 2014; Matcovitch-Natan et al., 2016). We find that *Tgm2* is broadly expressed from early embryonic microglia through adulthood. Consistent with our findings in freshly isolated postnatal microglia, *Tgm2* is expressed during critical windows of developmental myelination, and its expression in adult microglia is an order of magnitude higher. We also examined expression of *Tgm2* in the EAE murine model of multiple sclerosis, where its expression is not significantly different between unaffected and affected mice (Wlodarczyk et al., 2017). We have included these data as Figure 2—figure supplement 4 and Figure 4—figure supplement 1, respectively.